# Integrative neuromechanics of crawling in *D. melanogaster* larvae

Cengiz Pehlevan[1,2], Paolo Paoletti[3], L Mahadevan[4,5,6,7,8]*

[1]The Swartz Program in Theoretical Neuroscience, Harvard University, Cambridge, United States; [2]Simons Center for Data Analysis, Simons Foundation, New York, United States; [3]School of Engineering, The University of Liverpool, Liverpool, United Kingdom; [4]John A Paulson School of Engineering and Applied Sciences, Harvard University, Cambridge, United States; [5]Department of Organismic and Evolutionary Biology, Harvard University, Cambridge, United States; [6]Wyss Institute for Bioinspired Engineering, Harvard University, Cambridge, United States; [7]Kavli Institute for BioNano Science and Technology, Harvard University, Cambridge, United States; [8]Department of Physics, Harvard University, Cambridge, United States

**Abstract** Locomotion in an organism is a consequence of the coupled interaction between brain, body and environment. Motivated by qualitative observations and quantitative perturbations of crawling in *Drosophila melanogaster* larvae, we construct a minimal integrative mathematical model for its locomotion. Our model couples the excitation-inhibition circuits in the nervous system to force production in the muscles and body movement in a frictional environment, thence linking neural dynamics to body mechanics via sensory feedback in a heterogeneous environment. Our results explain the basic observed phenomenology of crawling with and without proprioception, and elucidate the stabilizing role that proprioception plays in producing a robust crawling phenotype in the presence of biological perturbations. More generally, our approach allows us to make testable predictions on the effect of changing body-environment interactions on crawling, and serves as a step in the development of hierarchical models linking cellular processes to behavior.

*For correspondence: lm@seas.harvard.edu

**Competing interests:** The authors declare that no competing interests exist.

## Introduction

A complete theory of locomotory behavior requires an integrative approach linking the nervous system in an organism to the body in which the nervous system lives and the environment that the body interacts with (*Pearson et al., 2006*; *Tytell et al., 2011*; *Chiel et al., 2009*). However, most studies focus on rhythmic gait and its maintenance in an organism driven by the presence of a central pattern generator (CPG) that drives coordinated motor activity (*Marder and Bucher, 2001*; *Ijspeert, 2008*). While the existence of a CPG has been validated in a variety of organisms (*Marder and Bucher, 2001*; *Grillner, 2006*) and exploited in artificial systems (*Ijspeert, 2008*; *Boxerbaum et al., 2012*), growing evidence suggests that sensory feedback plays an important role in maintaining robust and stable locomotion. Indeed recent studies on *C. elegans* (*Wen et al., 2012*; *Boyle et al., 2012*) focusing on local sensory feedback and proprioception show that these modalities suffice to modulate the locomotory pattern and explain gait transitions associated with undulatory swimming and crawling, without the need for a central pattern generator. This has led to recent attempts to include proprioceptive coupling and build an integrative theory of locomotion in examples such as anguilliform swimming in fish (*Ekeberg, 1993*; *Ekeberg and Grillner, 1999*), swimming in leech (*Cang and Friesen, 2002*), walking in insects (*Kukillaya et al., 2009*; *Proctor et al., 2010*;

*Proctor and Holmes, 2010*; *Holmes et al., 2006*) and humanoids (*Verdaasdonk et al., 2009*). However, the complexity of the brain-body-environment coupling in these organismal systems has been a substantial impediment in the use of models to make testable predictions on the biological mechanisms regulating locomotion.

Here, we consider the rectilinear crawling behavior of a model organism, the larva of the fruit fly, *D. melanogaster* (*Video 1*), which has increasingly become the focus of molecular, cellular, genetic and behavioral studies using a variety of experimental probes (*Suster and Bate, 2002*; *Fox et al., 2006*; *Hughes and Thomas, 2007*; *Crisp et al., 2008*; *Song et al., 2007*; *Lahiri et al., 2011*; *Inada et al., 2011*; *Berni et al., 2012*; *Heckscher et al., 2012*; *Crisp et al., 2011*; *Fushiki et al., 2013*; *Gjorgjieva et al., 2013*; *Vogelstein et al., 2014*; *Kohsaka et al., 2014*; *Itakura et al., 2015*; *Pulver et al., 2015*). The larva is a soft bodied cylindrical organism about 4 mm in length and about 800 µm in diameter in the third instar stage. It moves in a manner similar to other long soft-bodied creatures such as earthworms and leeches by exploiting the peristaltic propagation of muscular relaxation and contraction waves along their bodies to induce forward locomotion, or crawling (*Trueman, 1975*); despite biomechanical differences between these different organisms, the crawling gait seems to be a convergent strategy across species. The dynamical process that triggers, coordinates and maintains the propagation of such waves has attracted the attention of researchers for a century (*Garrey, 1915*). Early experimental efforts tried to understand the macroscopic mechanics of soft bodied animal locomotion (*Trueman, 1975*) by focusing on one of the underlying subsystems: body mechanics, muscular force production and neural dynamics. More recently, there is a growing realization that the coupling between the nervous system, the body and the substrate in the presence of sensory feedback plays a major role in development and maintenance of crawling gaits, and perhaps even in evolution (*Chiel et al., 2009*). For example, the locomotory behavior of *Manduca sexta* larvae, an organism where proprioceptive sensing displays a wide range of behaviors (*Simon and Trimmer, 2009*), is known to be dependent on substrate stiffness that modulates the role of external stimuli and on body deformation rate (*Lin and Trimmer, 2010*). In the *D. melanogaster* larva, there is strong evidence that proprioception plays as important a role as the CPG in generating coordinated motion. Eliminating proprioception with genetic (*Hughes and Thomas, 2007*) and optogenetic (*Inada et al., 2011*) methods leads to qualitative changes in the crawling gait: peristaltic waves have a period that is ≈ 4 times longer or may even stop propagating (*Song et al., 2007*), and body segments contract ≈ 2 times more (*Hughes and Thomas, 2007*). Experiments that block sensory input in the embryo show that crawling behavior still develops in the larva, but with longer peristaltic wave periods (*Suster and Bate, 2002*; *Fushiki et al., 2013*), providing further evidence for the importance of proprioception. Together, the classical and modern studies on locomotory physiology and the modern studies on the molecular and cellular subsystems in the larva suggest that it is an excellent biophysical testbed for an integrative theory that spans multiple scales.

In this paper, we present a mathematical model of crawling in *D. melanogaster* larvae that is guided by the anatomy and the kinematics of the gait of the organism. Our theory explicitly accounts for the mechanics of the passive deformable soft body, properties of the substrate on which the crawling occurs, active muscular forcing, neural dynamics and the interactions and feedbacks between these sub-systems. This allows us to reproduce the robust crawling gait that is consistent with experimental findings in first (*Heckscher et al., 2012*) and third (*Hughes and Thomas, 2007*) instar larvae. Furthermore, our model qualitatively and quantitatively captures the effects of a) optogenetic perturbations of neural activity (*Inada et al., 2011*; *Kohsaka et al., 2014*), and b) silencing proprioception with genetic (*Hughes and Thomas, 2007*) and optogenetic (*Inada et al., 2011*) methods.

Our integrated model also allows us to make specific experimentally testable predictions. In particular, by changing the strength of coupling between adjacent segments in the CPG, both in the absence and presence of proprioception, we show how proprioception increases the robustness of crawling. This leads to the prediction that there should be much more variability in crawling metrics among individuals with silenced or weakened proprioception. Furthermore, we predict that larvae could use the strength of CPG coupling as a means of controlling the speed of gait. Finally, we show that changing the frictional interactions of the organism with the substrate should yield observable effects on the efficiency of locomotion.

More broadly, our study also aims to provide a set of plausible scenarios for the biophysical mechanisms underlying crawling, by linking body mechanics, muscular forcing, neural dynamics, the properties of the substrate and their coupling, with natural implications for engineering applications.

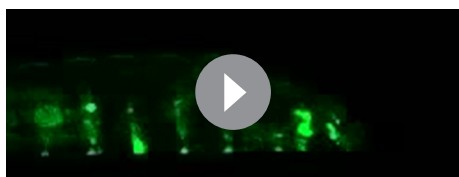

**Video 1.** GFP imaging of *Drosophila* larva forward crawling body segment and gut movements. Courtesy of Ellie Heckscher. See also reference (*Heckscher et al., 2012*). Original video is available at: https://www.youtube.com/watch?v=1d7zMYWLjLI

# Mathematical model

## Overview

We start with a broad overview of our model leaving aside the mathematical details and experimental justifications behind them in the rest of this section.

In *Figure 1*, we show our model of a *D. melanogaster* larva. Anatomically, the larva has 3 thoracic (T1-T3) and 8 abdominal (A1-A8) segments, *Figure 1A*. In addition, there is a head/mouth and a tail, but due to the lack of sharp boundaries in their musculature, experiments have not been able to distinguish the dynamics of the head motion from that of T1 and the tail from that of A8 (*Lahiri et al., 2011*; *Heckscher et al., 2012*). We therefore treat the head as a part of the T1 segment, and the tail as a part of the A8 segment. Moreover, here we focus on the simplest locomotory behavior associated with rectilinear motion along the anterior-posterior axis (*Hughes and Thomas, 2007*; *Heckscher et al., 2012*), thus ignoring the individual dynamics of hemisegments which move together in rectilinear crawling, and enumerate segments from 0 to 10 starting from the head.

The larva is modeled as a set of discrete, repetitive units, one unit for each body segment. The main features of each unit are illustrated in *Figure 1B*, while *Figure 1C* shows the collective dynamics of the multiple units. When stationary, each segment has length $L$, a parameter that sets the length scale in the model. The larva is soft-bodied and the elastic properties of body segments are approximated by a set of linear springs and dampers. A key parameter here is the stiffness of the springs that sets the scale of forces in the model. Each unit has a neural controller made of excitation-inhibition circuits in the Ventral Nerve Cord (VNC), which governs the reaction time of excitation in the neural controller. The excitatory neurons act also as motor neurons, and drive a muscle within the segment, which exerts a contractile force to the segment when activated. Crawling occurs on a substrate when the force rises above a threshold controlled by friction. Larvae lift segments off the ground when they contract (*Heckscher et al., 2012*) and hence control friction actively.

The contraction wave propagates through the body by a sequential activation of neural controllers in the VNC, leading to propagation through two channels (*Figure 1D*): 1) A proprioceptive channel that is mediated by 'Stretch receptors', which respond to changes in segmental length, get activated when a segment contracts beyond a threshold and send two excitatory signals. One of them feeds the excitatory neurons in the next anterior segment and propagates the neural activity. The other signal feeds to the inhibitory neurons in the same segment, leading to the inhibition of excitatory neurons that then causes contraction to be stopped. This model of proprioception is consistent with the 'mission accomplished' model of proprioception, proposed in (*Hughes and Thomas, 2007*). 2) A neural channel that is mediated by excitatory neurons in adjacent segments that are coupled from posterior to anterior direction.

Finally, our model also involves long range mechanical and neural/proprioceptive coupling between head and tail to trigger a new cycle of crawling as soon as the head starts moving. At the beginning of each crawling cycle, it has been observed that the head and the tail of a larva move concurrently in a motion called the 'visceral piston phase' of crawling (*Heckscher et al., 2012*; *Simon et al., 2010*), because the gut moves with the head and the tail, in advance of the surrounding body tissues. We model this behavior by forcing the displacement of head and tail segments to be the same. Once the peristaltic wave reaches the head, it propagates to the tail through assumed long range neural and proprioceptive couplings of T2 and A8 segments, and a new wave is initiated.

Our study builds on and extends a recent minimal model for crawling locomotion (*Paoletti and Mahadevan, 2014*) that shows how a local sensory feedback-based mechanism is capable of inducing rhythmic locomotion in soft bodied organisms by accounting for a fully coupled excitatory-inhibitory neural circuit, and the nonlinear frictional interaction with the substrate, while hewing close to experiments on the *D. melanogaster* larva.

It is useful to also contrast with a recent study (*Gjorgjieva et al., 2013*), which focused on the neural dynamics of the VNC and studied conditions under which activity propagates in the VNC. The

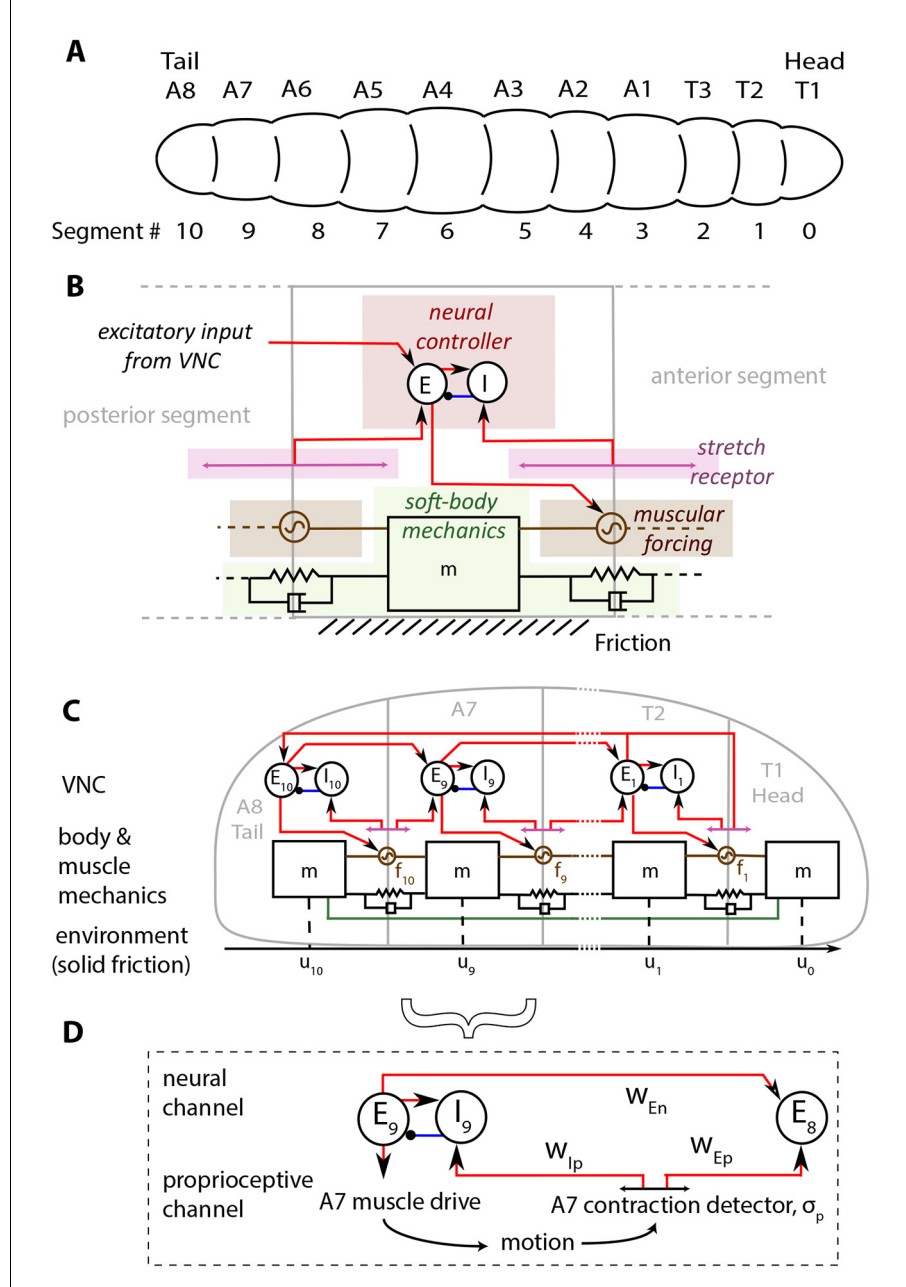

**Figure 1.** Schematic of the model. (**A**) *Drosophila* larvae have segmented bodies. (**B**) An overview of the model for one segment. Only input neural and proprioceptive signals are shown to the neural controller, its output is not shown. Shades mark subsystems. Red lines show excitatory neural synapses, blue denotes inhibitory synapses. (**C**) The larva body is modeled as a linear chain of masses, connected by damped linear springs. Head and tail segments are coupled mechanically, denoted by the green line. Each segment feels friction due to contact with the substrate. Body segments are actuated with muscular forces, $f_i$, that are excited by input from the larva VNC. VNC is modeled as a chain of excitatory, $E_i$, and inhibitory, $I_i$, neural populations. Self-coupling of populations are not shown. VNC gets proprioceptive input that signals contraction of a particular segment, shown by lines with arrows on both ends. (**D**) Segment-to-segment propagation of neural activity happens through neural ($w_{En}$) and proprioceptive ($w_{Ep}$) couplings. Detailed description and model equations are given in the Model section.

model differs from our neural model most importantly by including anterior-to-posterior couplings and intersegmental coupling from inhibitory to excitatory neurons, both of which were necessary to achieve bidirectional wave propagation in the VNC. The same study also considered a bilateral VNC model and examined its left-right synchronization properties. However, unlike the present study, this model did not include any mechanical, muscular and environmental coupling.

## Neural dynamics: excitation, inhibition and proprioceptive coupling

We model the neural dynamics of the ventral nerve chord (VNC) in terms of the classical Wilson-Cowan equations (*Wilson and Cowan, 1972*) for the activity of excitatory and inhibitory populations of neurons in a segment. This choice leads to a more realistic model than the single population phase oscillators used in (*Paoletti and Mahadevan, 2014*) and is thus similar to recent approaches that consider the purely neural aspects of the locomotory circuit (*Gjorgjieva et al., 2013*):

$$\tau_E \dot{E}_i = -E_i + \sigma_n[w_{EE}E_i + w_{EI}I_i + h_i^E - \hat{\theta}_E],$$
$$\tau_I \dot{I}_i = -I_i + \sigma_n[w_{IE}E_i + w_{II}I_i + h_i^I - \hat{\theta}_I], \ i = 1, \ldots, 10. \tag{1}$$

Here $E_i(t)$ and $I_i(t)$ are the activity levels of the excitatory and inhibitory neuron populations respectively, $w_{EE}, w_{EI}, w_{IE}$, and $w_{II}$ are the weights for the excitatory-excitatory, excitatory-inhibitory and inhibitory-inhibitory couplings, $\hat{\theta}_E$ and $\hat{\theta}_I$ are activation thresholds for the different neural populations, $\sigma_n[E] = 0.5 + 0.5\tanh(g_n E)$ is a sigmoid characterizing the switching threshold with $g_n$ the dimensionless gain, and $h_i^{E,I}$ refer to external inputs to these segmental populations. The external inputs take two forms: neural coupling that links to the neural populations in the neighboring posterior segment, and proprioceptive coupling that accounts for mechano-sensory feedback from the body to the VNC, shown in *Figure 1B,C and D*. A minimal mathematical description of this leads to the following dynamics of inputs $h_i^E(t), h_i^I(t)$ :

$$h_i^E = w_{En}E_{i+1} + w_{Ep}\sigma_p[u_{i+1} - u_i - \hat{u}], \ i = 1, \ldots, 9,$$
$$h_{10}^E = w_{En}E_1 + w_{Ep}\sigma_p[u_1 - u_0 - \hat{u}],$$
$$h_i^I = w_{Ip}\sigma_p[u_i - u_{i-1} - \hat{u}], \ i = 1, \ldots, 10. \tag{2}$$

For inputs to excitatory neurons, i.e. the first two equations, the first term on the right hand side corresponds to the input from the neural populations, which did not exist in (*Paoletti and Mahadevan, 2014*), while the last term corresponds to the proprioceptive input from the body. Here $w_{En}$ governs the strength of neural coupling, $w_{Ep}$ and $w_{Ip}$ the strengths of the proprioceptive couplings, $u_i(t)$ is the location of segment $i$, $\hat{u}$ being the segmental contraction threshold, and $\sigma_p[u] = 0.5 + 0.5\tanh(g_p u/L)$, where $g_p$ is the gain. The last *Equation in (2)* characterizes proprioceptive input to inhibitory neurons thresholded by $\hat{u}$. We have assumed that the response time of the stretch receptors is relatively fast compared to the dynamics of the VNC neural populations, and model the input from the stretch receptors as sigmoids, $\sigma_p$, weighted by parameters $w_{Ep}$ and $w_{Ip}$. Thus, when $w_{Ep} = 0$ and $w_{Ip} = 0$ there is no proprioception; as we will see, varying these parameters may be directly related to recent experimental manipulations (*Hughes and Thomas, 2007*; *Inada et al., 2011*).

Observations show that larvae can crawl without proprioceptive feedback (*Suster and Bate, 2002*; *Hughes and Thomas, 2007*; *Inada et al., 2011*) or input from the brain (*Berni et al., 2012*). Furthermore, the central nervous system, when isolated from the body, can still produce waves of neural activity propagating from posterior to anterior (*Pulver et al., 2015*). Therefore, the VNC should be able to propagate neural activity purely by segment-to-segment neural coupling; in our model this is achieved by introducing excitatory neural couplings from posterior to adjacent anterior segments, strengths of which are governed by the parameter $w_{En}$ (*Figure 1B,C and D*).

For the proprioceptive coupling we assume the 'mission accomplished' model (*Hughes and Thomas, 2007*; *Song et al., 2007*) (*Figure 1B,C and D*). In (*Hughes and Thomas, 2007*) silencing bipolar dendrite and class I multidendritic types of sensory neurons was shown to slow down frequency of peristaltic waves significantly. The 'mission accomplished' model (*Hughes and Thomas, 2007*) proposes that these neurons signal the VNC at the end of a successful contraction in a body segment. When a segment contracts relative to an adjacent anterior segment beyond a threshold,

an excitatory signal is sent to the next anterior VNC segment to initiate contraction. Simultaneously, activity in the current segment is suppressed by exciting the inhibitory population, a mechanism that was not accounted for in our earlier minimal model (*Paoletti and Mahadevan, 2014*).

We also assume neural and proprioceptive inputs from segment T2 to segment A8, as modeled by the second *Equation (1)*; these naturally lead to reinitiation of the crawling cycle. This again differs from (*Paoletti and Mahadevan, 2014*) and (*Gjorgjieva et al., 2013*) where reinitiation is achieved by external inputs. Such long range coupling, also used in stick insect models (*Daun-Gruhn and Tóth, 2011*), is plausible as neurons that extend across multiple segments have been observed in *Drosophila* larva VNC (*Schmid et al., 1999*). We note that this is not the only possible mechanism to reinitiate crawling cycles, and an alternative is discussed in the Appendix 1.

## Body mechanics: active muscular and passive tissue mechanics

At the anatomical level, we assume that each segment has mass $m$, length $L$ and is linked to its immediate neighbors by linear springs with stiffness $k$ and damping coefficient $c$, as shown in *Figure 1C* and similar to what is described in (*Paoletti and Mahadevan, 2014*). As we will see, this minimal mechanical model suffices to explain a range of experimental observations.

One cycle of larval forward crawling has two phases (*Heckscher et al., 2012*): (i) the 'visceral piston' phase (*Simon et al., 2010*), where the gut moves forward in advance of the surrounding tissues, concurrently with the head and the tail, followed by (ii) the wave phase, where the peristaltic wave propagates from posterior to anterior in the remaining segments. The mechanism underlying the tail-head coordination during the visceral piston phase is unknown, however the observation that the gut moves together with the head and the tail (*Heckscher et al., 2012*), is consistent with the suggestion that the gut mechanically couples the head and the tail and leads to visceral piston-like action as seen in other organisms such as the *Manduca sexta* larvae (*Simon et al., 2010*). We chose to minimally implement this coupling by linking the head and the tail with a rod, i.e. an infinitely stiff spring, thus enforcing a 'periodic' boundary condition that leads to concurrent head-tail movement. Softening the spring will introduce a small temporal delay and allows for changes in the length of the larva during peristalsis, but does not change the qualitative nature of our results. Our model ignores the frictional interactions of the gut with the body segments, however in the absence of experimental data to guide modeling such interactions, we choose to keep our model minimal. Another mechanism for enforcing such periodic boundary condition could be synchronized neural drive at the head and the tail segments, which requires more elaborate models that we explore in the Appendix 1.

Further, since the maggot and its segments move relatively slowly, we assume that inertial effects are negligible so that segmental forces are balanced by friction locally. Although experimental measurements of these forces do not yet exist, a simple estimate shows that this hypothesis is justified. The mass of a third instar larva is $\approx 10^{-3}$ g and the acceleration of the larvae is on the order of $\approx 10^{-3}$ m/s², which leads to an inertial force of $\approx 10^{-6}$ gm/s². In an experimental study (*Wallace, 1969*), friction forces to draw a glass fiber of the size of a small nematode on an agar surface was measured to be $\approx 5 \cdot 10^{-2}$ gm/s², justifying our approximation.

Collecting all these arguments together, the displacement of individual segments $u_i(t)$ are governed by the equations:

$$0 = k(u_{i-1} - 2u_i + u_{i+1}) + c(\dot{u}_{i-1} - 2\dot{u}_i + \dot{u}_{i+1}) + f_i - f_{i+1} - F_i^f, \qquad i = 1, \dots, 9$$
$$0 = k(u_9 - u_{10} + u_1 - u_0) + c(\dot{u}_9 - \dot{u}_{10} + \dot{u}_1 - \dot{u}_0) + f_{10} - f_1 - F_{10}^f - F_0^f,$$
$$0 = u_{10} - u_0 - 10L, \tag{3}$$

where $F_i^f$ is the frictional force on the body segments, and $f_i$ are the muscular forces. Here, the penultimate equation characterizes the mechanics of the first segment, while the last equation describes our enforced periodic boundary condition to model the concurrent head-tail motion in the visceral piston phase, a boundary condition that was not used in our earlier model (*Paoletti and Mahadevan, 2014*) that required an external periodic excitation signal to achieve sustained crawling.

Muscles in each segment provide contractile forces in the anterior-posterior axis necessary for locomotion (*Heckscher et al. (2012)*). They are activated by excitatory input from the neurons of the corresponding segment of the VNC. This is consistent with the observation that the propagation of contraction waves can be temporarily stopped by locally inhibiting the motor neurons in one

segment (*Inada et al. (2011)*). As detailed studies of the muscular dynamics are not available in the larva, we model this behavior using a simple first-order dynamical law for the muscular force

$$\tau_f \dot{f}_i = -f_i + f_{\max}\sigma_f[E_i - \hat{E}], \qquad i = 1, \ldots, 10, \tag{4}$$

where $f_{\max}$ is the maximum force exerted by muscles, $\sigma_f[E] = 0.5 + 0.5\tanh(g_f E)$, where $g_f$ is the gain and $\hat{E}$ is a threshold for muscle activation.

Due to the assumed head-tail coupling, contraction of the tail segment leads to stretching of the head segment driven by $f_{10}$, consistent with the observation in (*Heckscher et al., 2012*), that tail muscles contract in the visceral piston phase. In (*Heckscher et al., 2012*), it was speculated that tail contraction provides both a moving of the tail forward and pushing of the gut forward. Our model could be interpreted as adding to this speculation that the push gets transferred to the head by the gut and causes its motion. It is possible that motion in head segment is not totally passive, but mediated actively by contraction of muscles in the head. However, experimental evidence on this issue is not decisive (*Heckscher et al., 2012*) and we do not consider this scenario here.

## Environmental mechanics: substrate frictional coupling

Directed rectilinear locomotion is a consequence of body contraction coupled to the anisotropic and inhomogeneous friction of the body relative to the substrate. Frictional inhomogeneity arises as segments are lifted off the ground when they contract (*Heckscher et al., 2012*), allowing them to slip and providing the organism active control of friction. Indeed, activity of the muscles that coordinate segment lifting are synchronized with the activity of muscles that provide contractile forces in the segment (*Heckscher et al., 2012*). Consistent with this, we assume that the frictional resistance $F_i^f$ vanishes when $f_i$ is above a threshold, i.e.

$$F_i^f = F_{\max}\operatorname{sign}(\dot{u}_i)\sigma_F[\hat{f} - f_i], \qquad i = 1, \ldots, 10,$$
$$F_0^f = F_{\max}\operatorname{sign}(\dot{u}_0)\sigma_F[\hat{f} - f_{10}], \tag{5}$$

where $F_{\max}$ is the maximum frictional force, $\sigma_F[f] = 0.5 + 0.5\tanh(g_F f/kL)$, where $g_F$ is the gain, and $\hat{f}$ is the threshold segmental muscular force associated with segment lifting and the resulting vanishing of friction. This nonlinear frictional interaction is different from that in our earlier minimal model (*Paoletti and Mahadevan, 2014*) where the strength of friction was dependent on the direction of motion. Again, the last equation describes the head-tail coupling. We note that the frictional interaction of the body with the substrate decouples segments far from each other both mechanically (due to inhomogeneous deformation) and neurally (due to inhomogeneous proprioception). Manipulating the body-substrate coupling by changing $F_{\max}/kL$ allows us to make experimentally testable predictions for gait changes.

*Equations (1–5)* characterize the coupled neuromechanics of the larva linking the brain, body and environment by incorporating the neural dynamics that induces muscle contraction, the passive and active mechanics of the body, and the frictional interaction with the substrate on which the maggot moves, and the various interactions between these subsystems. Together with initial conditions, this completes the formulation of the problem. Our differential equations have strong nonlinearities associated with the sigmoids, which makes them numerically stiff. Introducing small inertial contributions for the segments allow us to use explicit numerical integration schemes encoded in MATLAB, although our results are robust with respect to changes in this parameter (see Appendix 2). We note that our numerical solution method differs from that used in (*Paoletti and Mahadevan (2014)*), where a continuum limit (when the number of segments is large) was taken first to derive a partial differential equation that was then solved numerically.

## Parameters

Our model is characterized by a number of dimensionless parameters that are given by the dimensionless damping $c\tau_E/k$, the scaled maximum frictional force $F_{\max}/kL$, the scaled maximum muscular force $f_{\max}/kL$, the scaled threshold muscular force that causes segment lifting $\hat{f}/kL$, the activation thresholds for neural populations $\hat{\theta}_E$ and $\hat{\theta}_I$, the activation threshold for muscular forces $\hat{E}$, the scaled segment displacement $\hat{u}/L$, neural network weights $w_{EE}$, $w_{EI}$, $w_{IE}$, $w_{II}$, $w_{En}$, $w_{Ep}$, $w_{Ip}$, the gains $g_n$, $g_f$,

$g_p$, $g_F$ and the scaled muscular relaxation time scale and inhibitory time scale $\tau_f/\tau_E$, $\tau_I/\tau_E$. All our results are reported in units of segment length $L$, neural excitation time constant $\tau_E$ and body stiffness $k$. To initiate crawling, we apply a rectangular pulse of height 0.61 for a duration of $10\tau_E$ to the appropriate segment's excitatory neuron.

The dimensionless parameters in the problem were adjusted to approximate the quantitative results reported in (*Heckscher et al., 2012*; *Hughes and Thomas, 2007*), and reproduce the effects of optogenetic perturbations of neural activity in the VNC (*Inada et al., 2011*; *Kohsaka et al., 2014*) (*Table 1*). First, the VNC model parameters were chosen so that the VNC by itself could produce repeated propagation of activity. We made sure that the VNC was configured much below the maximum excitation it can carry, to make room for the additional proprioceptive input when the full model is put together. Next, the scaled muscular force was chosen so that $f_{\max}/kL \le 1$ for stability, and to allow for the large contractions observed in crawling without proprioception. Finally, the scaled friction force was chosen so that $F_{\max}/kL >> 1$ to avoid slippage. Together, this allowed us to find a stable crawling solution that matches experimental results (*Heckscher et al., 2012*; *Hughes and Thomas, 2007*; *Inada et al., 2011*; *Kohsaka et al., 2014*).

## Results

We start with a quantitative description of our model results and their comparison to experiments before turning to make testable predictions.

### Experimental validation of the model

#### The model produces sustained crawling with metrics matching experiments

Crawling is initiated with a short excitatory pulse to the excitatory neural population in the most posterior segment. We do not model the source of this initiation command; it could be, for instance, a descending signal from the brain initiating forward movement or sensory feedback from the tail initiating an escape. This yields a sustained crawling gait shown in *Figure 2*, where a kymograph of body segments of the larva (*Figure 2A*) as well as the corresponding muscular (*Figure 2B*) and neural activity (*Figure 2C*) are shown (also see *Video 2*). We see that head and tail segments move together in the visceral piston phase (*Heckscher et al., 2012*; *Simon et al., 2010*), followed by a peristaltic wave with neural activity leading in phase, followed by muscular and contraction activity (*Figure 2*), propagating from posterior to anterior segments. Crawling can be stopped by shutting down the activity of excitatory neurons in the most posterior segment by an inhibitory input (see also section 3.1.2).

Our simulations show that the larva produces approximately 0.04 waves per unit time associated with the relaxation of the excitatory neurons $\tau_E$, where waves start when the tail moves off the ground, leading to a larval speed $\approx 0.04L/\tau_E$. Hughes and Thomas (*Hughes and Thomas, 2007*) found that third instar larvae of typical length $10L \approx 4$ mm (estimated from *Figure 1* of (*Hughes and Thomas, 2007*)) produce $\approx 1.5$ waves/s. Using this latter number, we can estimate the time scale: $\tau_E \approx 25$ ms, a reasonable time constant for activity of neural populations. Combining the time scale estimate and the length of the third instar larvae, our model predicts a speed of $\approx 0.5$ mm/s, comparable with the observed speed of $\approx 1$ mm/s (estimated from Figure 1 of *Hughes and Thomas, [2007]*). In another experiment, for first instar larvae with typical length $10L \approx 600$ μm (estimated

**Table 1.** Dimensionless parameters of our model and their default values used in numerical simulations.

| $c\tau_E/k$ | 3.5 | $f_{\max}/kL$ | 5/6 | $\tau_f/\tau_E$ | 0.4 |
|---|---|---|---|---|---|
| $F_{\max}/kL$ | 25/3 | $\hat{f}/kL$ | 5/12 | $\tau_I/\tau_E$ | 3 |
| $w_{EE}$ | 1 | $w_{EI}$ | -2 | $w_{IE}$ | 0.6 |
| $w_{II}$ | 0 | $w_{En}$ | 0.6 | $w_{Ep}$ | 1.95 |
| $w_{Ip}$ | 1.95 | $\hat{E}$ | 0.4 | $\hat{\theta}_{E,I}$ | 0.6 |
| $\hat{u}/L$ | -17/18 | $g_n$ | 40000 | $g_{f,p,F}$ | 1000 |

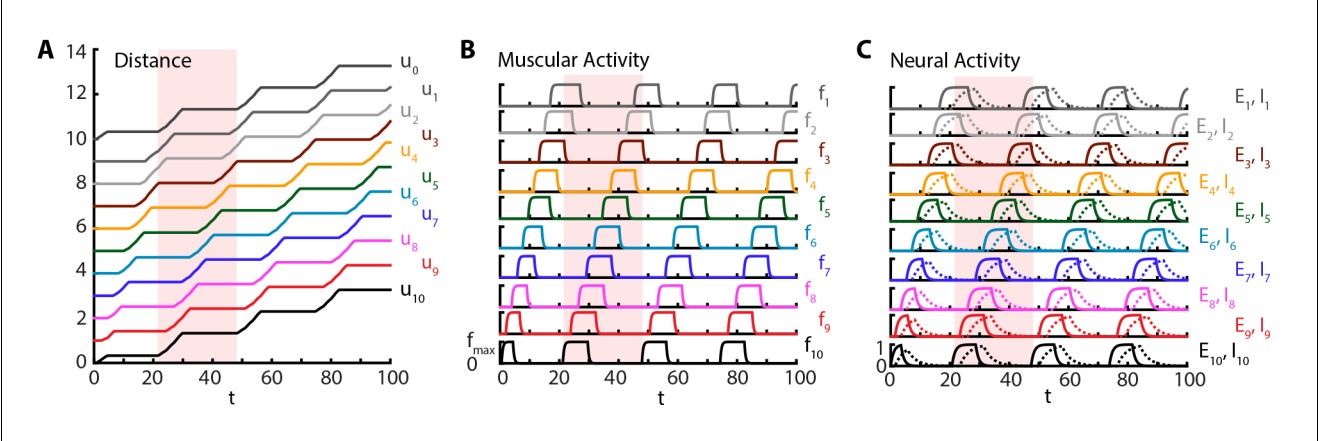

**Figure 2.** Sustained crawling. Time is in units of $\tau_E$. Model is simulated for $500\tau_E$, only first $100\tau_E$ is shown. All simulations were based on *Equations (1–5)*, with parameter values as specified in the *Table 1*. Shade shows the duration of an example peristaltic wave. (**A**) Kymograph of body segments. Distance is measured in units of *L*. (**B**) Muscular forces in segments. Same color code is used as in the **A**). (**C**) Neural activity in segments. Solid line denotes excitatory neuron population activity, and dashed lines denote inhibitory population. Same color code is used as in the **A**)

from Figure 1 of *Heckscher et al. [2012]*), Heckscher et. al. *Heckscher et al. (2012)* give a range of $0.5 - 1.5$ waves produced per second, which suggests the neural time scale range $\tau_E \approx 25 - 75$ ms, again a reasonable time constant range. Combining the time scale estimate and the length of the first instar larvae, our model predicts a speed range of $\approx 30 - 90$ μm/s, in agreement with the range reported in (*Heckscher et al., 2012*): $\approx 40 - 125$ μm/s. Our simulations show that typically (median across time steps), three segments are off the ground, defined by the number of segments in which $f_i$ is above $\hat{f}$ and thus friction vanishes, consistent with observations (*Heckscher et al., 2012*). Furthermore, we find that peak segmental contraction, averaged over segments and waves, is $\approx 30\%$, consistent with observations (*Hughes and Thomas, 2007*).

## The model reproduces the effects of optogenetic VNC perturbations

Our model shows that normal segment-to-segment propagation of activity arises through two different but coordinated channels, the posterior-to-anterior neural coupling between excitatory populations in adjacent segments, and via proprioceptive coupling (*Figure 1C and D*). Experimental perturbations of these channels are known to change the crawling modalities.

For example, recent advances in optogenetics have allowed for targeted manipulations of specific neuron types in the VNC (*Inada et al., 2011*; *Kohsaka et al., 2014*; *Itakura et al., 2015*). When segment-to-segment propagation was perturbed with optogenetic inhibition of motor neurons in a VNC segment (*Inada et al., 2011*), crawling stopped when the wave reached that segment. Conversely, when the inhibition is removed after up to $10$ s, the larva resumed crawling from the same segment. To see whether this observation is reproduced in our model, we performed an acute shutdown of the excitatory population in a segment, modeling the effect of optogenetic inhibition on motor neurons. *Figure 3A–C* and *Video 3* show the results of this simulation. Crawling is stopped at segment A6 ($u_8$) by setting $E_8 = 0$ when $t \in [65, 95]$. We note that crawling continues in this time frame until it reaches segment A6, consistent with that observed experimentally (Figure 7 of (*Inada et al., 2011*)).

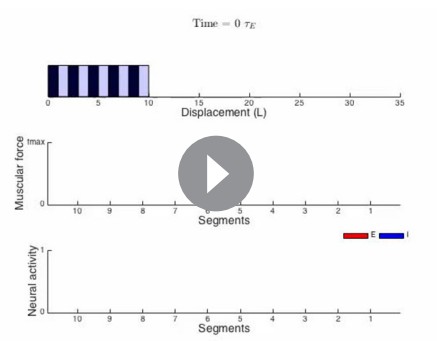

**Video 2.** Crawling simulations of full model larva. To illustrate the changes in friction, body segments are drawn off the ground when $f_i$ exceeds $\hat{f}$ in *Equation (5)*.

Additionally, even after crawling stops, contraction of the body segment $A7$ keeps producing proprioceptive input to $E_8$ and $I_9$. This can be seen by the sustained activity of $I_9$ during stopping, while the activity of $E_9$ dies out. When the optogenetic inhibition is removed, allowing $E_8$ to evolve according to its normal dynamics (*Equation (1)*), proprioceptive input drives $E_8$ to firing and the larva resumes crawling from A6.

In our model (*Figure 3*), during a stop, the memory of crawling phase is kept in the contraction of the body and the proprioceptive channel which drives the proprioceptive activity. In Inada *et. al.*'s experiments (*Inada et al., 2011*), it is possible to resume crawling even after the neighboring posterior segment to the one being inhibited completes a contraction and relaxes back to its equilibrium length (Figure 7 of (*Inada et al., 2011*)). Hence, Inada *et. al.* propose that the memory is held in the VNC. At first this might seem to contradict our result that the memory is kept in the body and the proprioceptive channel, but, although our model does not include such mechanism, it is possible for the proprioceptive neurons to exhibit self-sustained activity and be the location of memory. We also want to point that in (*Inada et al., 2011*) the memory can last as long as 10 s. Whether in the VNC or in the proprioceptive neurons, the mechanism that creates such long time intervals from neural time scales of a few tens of milliseconds is an open problem.

More recent experimental studies (*Kohsaka et al., 2014*; *Itakura et al., 2015*) identified premotor inhibitory neuron populations that play a role in locomotion. Specifically, during forward crawling, period-positive median segmental interneurons (PMSIs) were found to be activated slightly later than the motor neurons in the same segment (*Kohsaka et al., 2014*) and their optogenetic activation leads to inhibition of motor neuron activity locally, arresting peristaltic crawling. These observations match nicely with the dynamics of the inhibitory neurons in our model, which are activated slightly later than the excitatory neurons in the same segment (*Figure 2C*). Further, increasing their activity above what is normally seen during normal crawling led to the local arrest of crawling. In *Figure 3D–F* and *Video 4*, we show the results of a simulation where crawling is stopped at segment A6 ($u_8$) by setting $I_8 = 1$, the maximum inhibitory population activity in our model, over the time period $t \in [65, 95]$. Increased inhibitory neuron activity prevents the excitatory neurons from becoming active and stops the peristaltic wave. This causes the behavior of the model to be similar to the previous case where $E_8$ was shut down, except that here $I_8$ is active in the time frame of perturbation. When $I_8$ is left to evolve according to to its normal dynamics (*Equation (1)*), proprioceptive input drives $E_8$ to fire and larva resumes crawling from A6.

In the optogenetic inhibition experiments of *Kohsaka et al. (2014)*, when the activation of PMSIs is removed, the peristaltic wave did not continue from the inhibited segment, but a new wave from the posterior end started. This is in contrast to our model, where crawling continues from the inhibited segment. Other observations on PMSIs (*Kohsaka et al., 2014*) that remain to be incorporated into future models are: when inhibited using genetic and optogenetic methods, speed of peristalsis greatly decreased, duration of motor neuron bursting and muscle contraction increased however degree of segmental contraction did not change.

## The model reproduces effects of silencing proprioception

In another set of experiments, it was shown that silencing proprioceptive feedback to VNC using genetic (*Hughes and Thomas, 2007*; *Suster and Bate, 2002*) and optogenetic (*Inada et al., 2011*) methods leads to an increase in peak segmental contraction from $\approx 30\%$ to $\approx 65\%$ (*Hughes and Thomas, 2007*), and a reduction in rate to approximately one fourth (*Inada et al., 2011*) and to one tenth (*Hughes and Thomas, 2007*), and reduced speed (*Hughes and Thomas, 2007*). To understand this, we implemented a purely neural coupling in our model by setting proprioceptive couplings $w_{Ep} = 0$ and $w_{Ip} = 0$. With purely neural coupling, the model still produces sustained crawling but with a qualitative change in the crawling pattern (*Figure 4* and *Video 5*). Only a single segment is off-the-ground at a time, while peak segmental contraction, averaged over all segments, increases to $\approx 65\%$ of segmental length ($\approx 70\%$ for only A3-A4 segmental distance, *Figure 4E*) the larva produces a reduced rate of $\approx 0.01$ waves per $\tau_E$ and moves with speed $\approx 0.01 L/\tau_E$ (*Figure 4A and D*), in agreement with experiments. Furthermore, we see that inhibitory neural dynamics show a significant modulation, due to the removal of excitatory proprioceptive input to these populations, and peak activity of inhibitory population is reduced and delayed, as shown in *Figure 4C*. However, the phases

**Figure 3.** Perturbations of VNC. Top row: $E_8$ is inhibited for $t \in [65, 95]$, shown in purple. Bottom row: $I_8$ is maximally excited in the range $t \in [65, 95]$, shown in purple. Color code and parameters are the same as *Figure 1*. (**A**) and (**D**) Kymograph of body segments. (**B**) and (**E**) Muscular forces in segments. (**C**) and (**F**) Neural activity in segments. We note that in these simulations, crawling is started at $t = 0$ with an excitatory pulse applied to $E_6$ of the stationary larva, demonstrating that our model allows for the peristaltic wave to start at any segment.

of muscular and neural activity within a wave do not change, but the contraction phase is slightly earlier (*Figure 4F*).

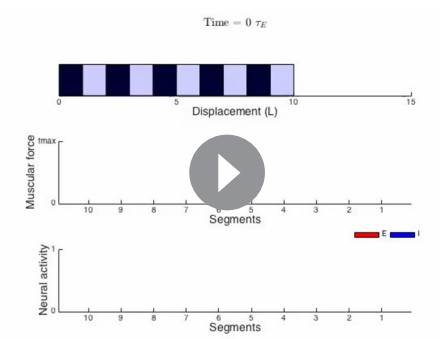

**Video 3.** Crawling simulations of full model larva. Cycle is started at A4 and temporarily stopped at A6 by inhibiting the excitatory population.

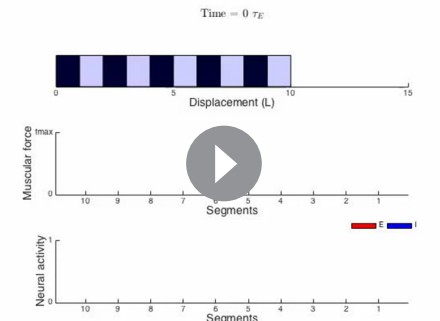

**Video 4.** Crawling simulations of full model larva. Cycle is started at A4 and temporarily stopped at A6 by exciting the inhibitory population.

## Predictions

### Role of proprioceptive coupling

We tried the converse of silencing proprioception, and implemented a purely proprioceptive coupling with no neural coupling by setting the posterior-to-anterior neural coupling $w_{En} = 0$. In this case, we find that our model produces sustained crawling almost identical to what is shown in *Figure 2* (see also below) where both channels are intact. This shows that a spatially coordinated CPG is not necessary for crawling, as was also suggested and shown in the simpler model put forward in (*Paoletti and Mahadevan, 2014*). While this might not be biologically realistic in an adult, our model shows that this plausible arrangement might be relevant in engineering coordinated locomotion, and even of some interest in an evolutionary or developmental setting.

The relative insensitivity of our model to variations in the parameters that describe neural coupling and the qualitative changes that result from silencing proprioception led us to investigate the stabilizing effect of proprioception in more detail. To this end, we varied posterior-to-anterior neural coupling, $w_{En}$, and investigated its effect when proprioception was silenced or intact. In *Figure 5*, we quantify various metrics characterizing the dynamics of the segments. As a function of $w_{En}$, *Figure 5A* shows the frequency of peristaltic waves which increases with $w_{En}$: stronger neural coupling leads to faster propagation of peristaltic wave. While higher frequency of peristaltic waves leads to faster locomotion as shown in *Figure 5B*, the relationship between speed and wave frequency is not linear as the size of step taken per peristaltic wave also increases with $w_{En}$, see *Figure 5C*. *Figure 5D* shows that peak contraction falls as $w_{En}$ increases and *Figure 5E* shows that the number of simultaneously off-ground segments increases as $w_{En}$ increases: faster peristaltic waves leave less time for segments to contract.

This manipulation suggests that proprioception increases robustness of locomotory behavior in two distinct ways. First, there is a wider range of neural coupling over which sustained crawling can be achieved. When proprioception is silenced, there is a minimum value of $w_{En}$ below which crawling is not possible, due to neural coupling being too weak to excite the neural population in the next segment (*Figure 5*). Proprioception provides the extra excitation that allows for sustained crawling till the neural coupling weight $w_{En} = 0$. Second, in all metrics that we used to characterize crawling, the observed changes as a function of $w_{En}$ was smaller when proprioception was intact (*Figure 5*). Thus, proprioception has a stabilizing effect on crawling. *Gjorgjieva et al. (2013)* came to a similar conclusion with their purely neural model.

In *Figure 5* all metrics of crawling with and without proprioception show a cross-over around $w_{En} \approx 0.65$, which also sets the threshold below which sustained crawling in the full model shows no dependence on $w_{En}$. To understand these further, we investigate the propagation delays through both the neural and proprioceptive channels. In *Figure 5F*, we plot three quantities: 1) the time it takes for excitatory neurons in neighboring segments to cross $\hat{E}$ (threshold to activate muscular forces) in the presence of proprioception (blue line), which quantifies segment-to-segment signal propagation delay, 2) the time for supra-threshold activation of excitatory neurons without proprioception (dashed red line), which quantifies the propagation delay through the neural channel, and 3) the time it takes for the excitatory population in a segment to cross $\hat{E}$ and the turning on of proprioceptive signal in that population (black line), which quantifies the propagation delay through the proprioceptive channel. We see that segment-to-segment signal propagation in the full model follows the faster channel, while in the proprioception silenced model it always follows the neural channel. Proprioceptive propagation delay becomes comparable to neural propagation delay around $w_{En} \approx 0.65$, the threshold when the metrics of crawling with and without proprioception cross-over. Below this threshold, proprioception is faster, explaining why the full model is insensitive to $w_{En}$. Above this threshold, neural propagation is faster, but proprioception still has a stabilizing effect in this regime because of its inhibitory effect on the neural activity in the moving segment, neutralizing strong excitation. Furthermore, the proprioceptive channel gets slower beyond the cross-over point: it takes longer for a segment to contract to the proprioceptive threshold point. This is consistent with decreased peak contraction and increased number of simultaneously moving segments.

Our results with varied proprioceptive coupling lead us to make the following predictions:

1. Crawling without any direct segment-to-segment coupling in the VNC is possible by segment-to-segment transmission of activity through the proprioceptive channel (*Figure 5*).

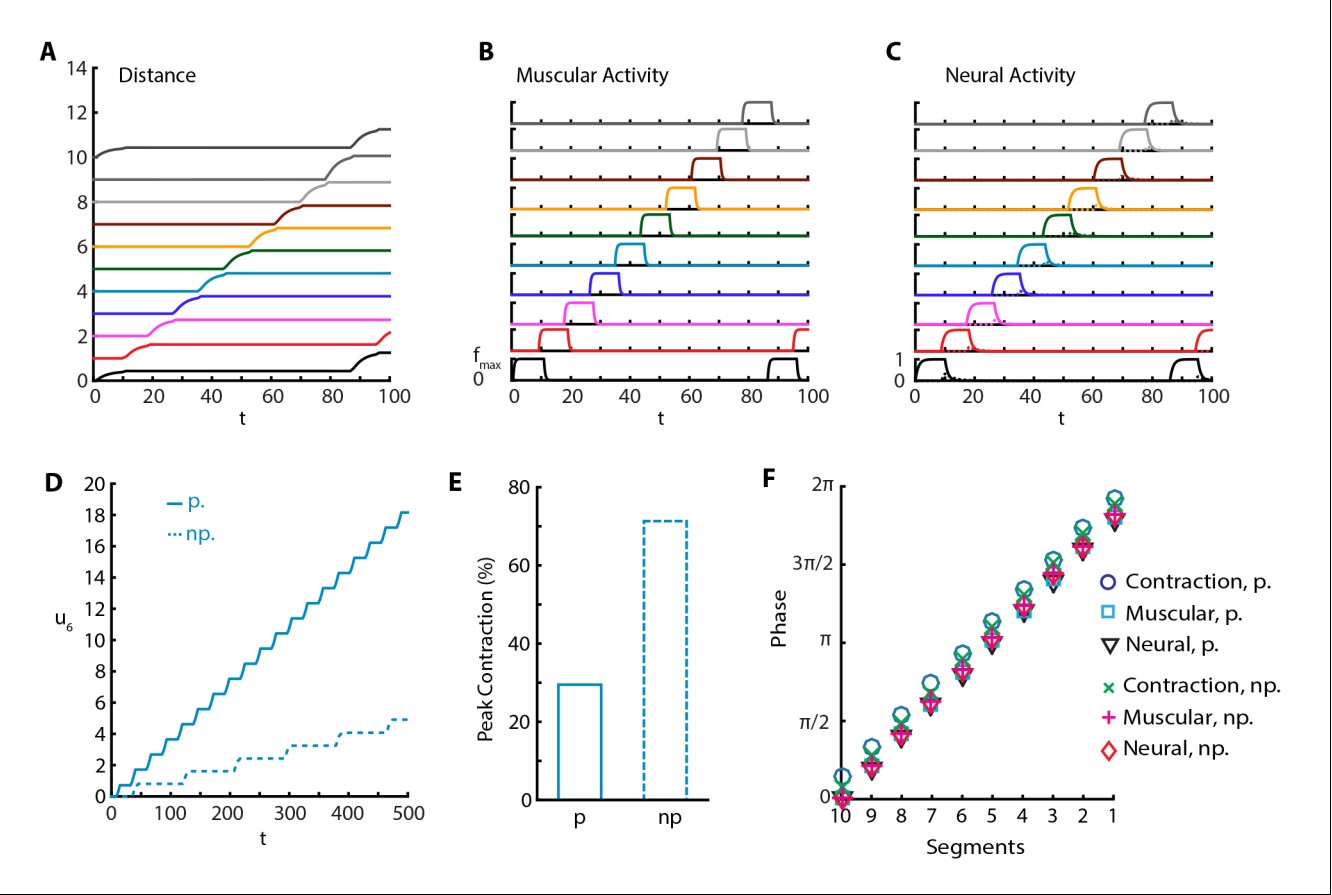

**Figure 4.** Sustained crawling without proprioception. The absence of proprioception induces slower crawling. In the lower panels, "p." denotes crawling with proprioception, 'np.' denotes crawling without proprioception. (**A**) Kymograph of body segments. (**B**) Muscular forces in segments. (**C**) Neural activity in segments. (**D**) Kymograph of A4 segment ($u_6$) in the full model vs. the model without proprioceptive feedback. (**E**) Peak contraction of A3-A4 ($u_5 - u_6$) segmental distance averaged over waves. Here, we plot only one segment interval as in (*Hughes and Thomas, 2007*) to ease comparison. (**F**) Phases of (excitatory) neural, muscular and contraction activity in different segments within a wave, with respect to excitatory neuron activity in tail segment. To calculate phases, first Discrete Fourier Transform of the relevant signal is obtained. Phase is the negative complex phase of the fundamental frequency. Contraction of a segment is the difference between its and next anterior neighbor segment's center of mass displacements.

2. Variation in parameters can be associated with variations across individuals in a population. Then, there should be much more variability in crawling metrics among individuals with silenced or weakened proprioception, a testable prediction using existing genetic tools (*Hughes and Thomas, 2007; Inada et al., 2011*).

3. Our model predicts that in the presence of proprioception, increase in neural coupling leads to higher speeds, as was also observed in (*Gjorgjieva et al., 2013*) in a purely neural model. In an analogous scenario in stick insects, experiments suggest that slow, steady-walking is mainly coordinated by local proprioceptive signals but faster motion is driven by increased neural coupling between legs (*Büschges, 2012*).

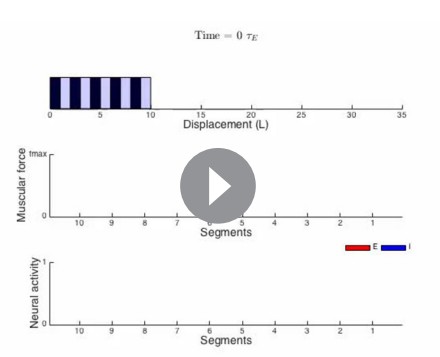

**Video 5.** Crawling simulations of larva with silenced proprioception.

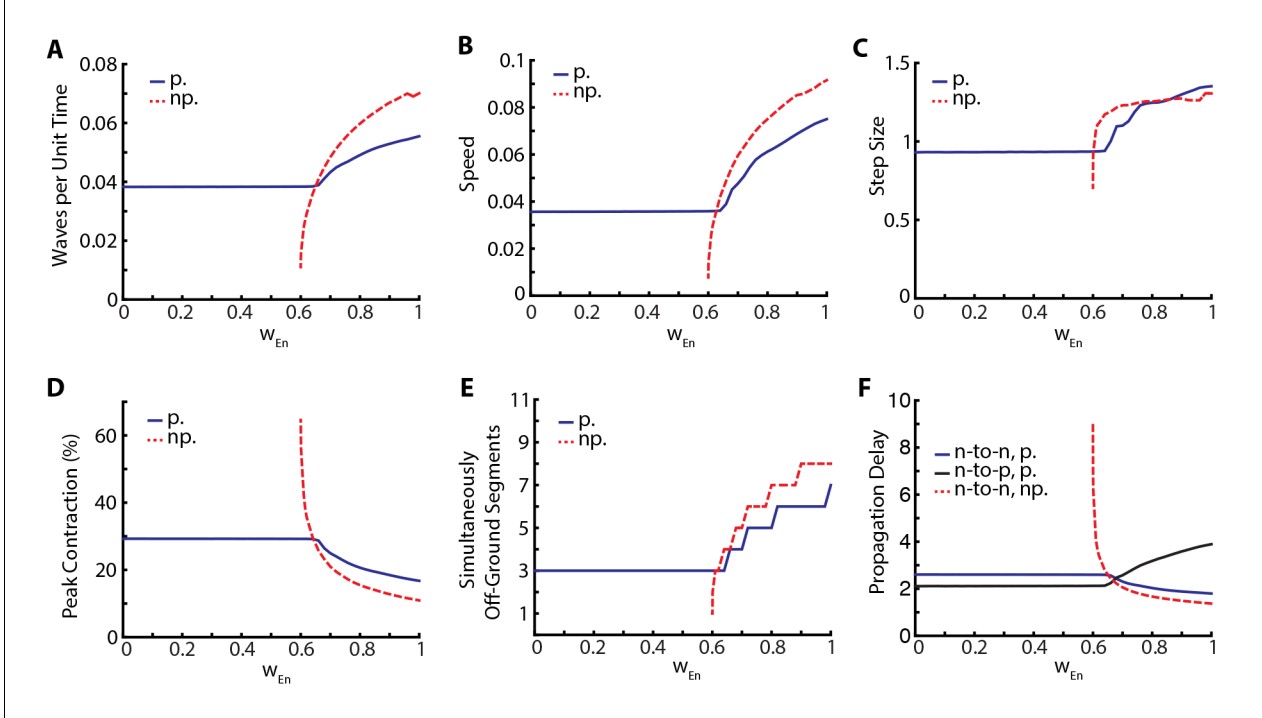

**Figure 5.** Proprioception increases robustness of crawling. Crawling metrics as a function of $w_{En}$. In dashed-red, we show the results from the model without proprioceptive feedback (np.), while in blue and black, we show the results with proprioception (p.) based on **Equation (1–5)** with parameter values as described in the **Table 1**. (**A**) Peristaltic waves per unit time ($\tau_E$). (**B**) Speed of crawling in units of $L/\tau_E$. (**C**) Step size, defined as the tail displacement divided by number of waves. It is measured in units of (L) **D**) Peak contraction in a segment, averaged over segments and waves. E) Number of simultaneously off-ground segments. This number is calculated at each time step, and the median number across time steps are plotted. (**F**) Propagation delays through neural and proprioceptive channels (see text for a detailed discussion) are shown. 'n-to-n' denotes neuron propagation delay. 'n-to-p' denotes proprioceptive propagation delay.

Then, our model suggests that the larva might modulate the strength of neural coupling to control speed. This suggestion brings accompanying predictions which can be read directly from **Figure 5**, e.g. faster speed comes with decreased peak contraction and a higher number of simultaneously off-ground segments.

## Role of body-substrate frictional coupling

To quantify the effect of the environment on the locomotion, we varied the maximum scaled frictional force, $F_{max}/kL$ between body and the substrate. Our results are shown in **Figure 6** and **Video 6**. Our main observation is that metrics of crawling are robust until friction falls below a threshold ($F_{max}/kL \sim 0.4$), when muscular forces become stronger than friction, leading to slippage. At this point the number of moving segments start differing from number of off-ground segments (**Figure 6F**). An example kymograph from the slipping regime ($F_{max}/kL = 0.005$), and corresponding muscular and neural activities are shown in **Figure 6A** and **Video 6**. Below the threshold, number of peristaltic waves per unit time drops (**Figure 6B**), as is confirmed by increased segment-to-segment propagation delay in the VNC (**Figure 6G**). Thus we see an effect on neural propagation due to a change in the mechanical interaction with the substrate, clearly showing how we cannot ignore the triad of nervous system-body-substrate in continuous conversation with each other. The speed of the larva (**Figure 6C**) also follows an interesting trend; with decreasing friction, speed first increases due to larger step sizes (**Figure 6D**), even though the frequency of peristaltic waves decreases (**Figure 6B**), but eventually saturates. Peak contraction of segments increase (**Figure 6E**) with decreased friction. These effects on behavior are clear testable predictions of our model.

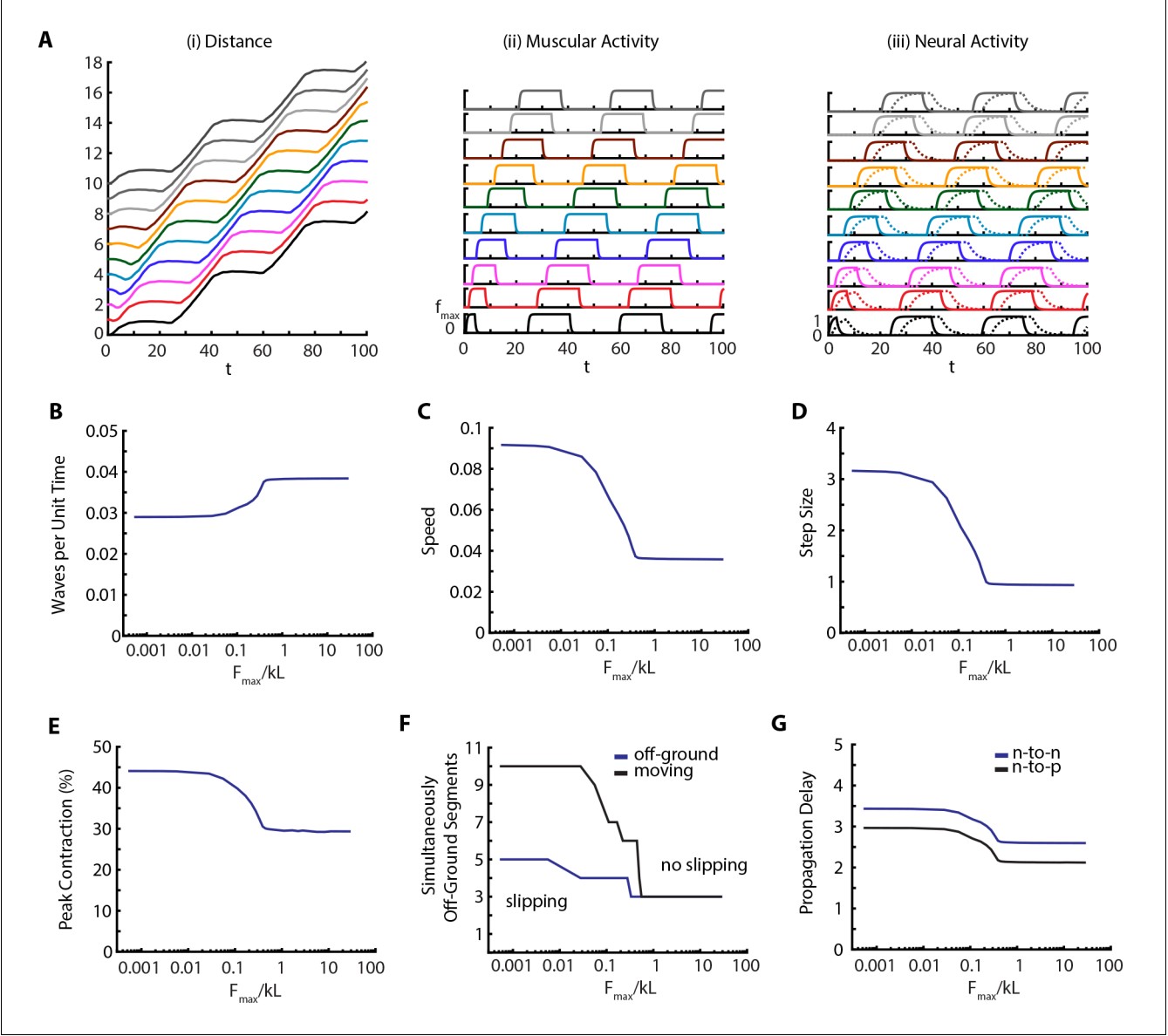

**Figure 6.** Decreasing friction leads to slippage. (**A**) Kymograph, muscular forces and neural activity as a function of time for $F_{max}/kL = 0.005$. See **Figure 2** and text for explanation of plots. (**B,C,D,E**) Crawling metrics as a function of $F_{max}/kL$. See **Figure 5** and text for explanation of plots.

## Discussion

Our theoretical model for the forward crawling of a *D. melanogaster* larva incorporates the coupled mechanics of the soft body, the neural dynamics of VNC and frictional interactions with the substrate, complementing earlier isolated studies of these sub-systems. It produces a robust, sustained crawling gait with metrics consistent with experimental findings (*Heckscher et al., 2012*; *Hughes and Thomas, 2007*), and furthermore, can reproduce qualitative and quantitative changes in crawling gait due to perturbations in proprioceptive (*Hughes and Thomas, 2007*; *Inada et al., 2011*) and neural (*Inada et al., 2011*; *Kohsaka et al., 2014*) channels of segment-to-segment wave propagation.

A surprising finding of our model is its ability to produce sustained crawling with purely proprioceptive coupling between segments, a scenario which was first suggested by us for a very general model of crawling (*Paoletti and Mahadevan, 2014*). While both in our model and in experiments (*Hughes and Thomas, 2007*; *Heckscher et al., 2012*), the larvae can crawl without any

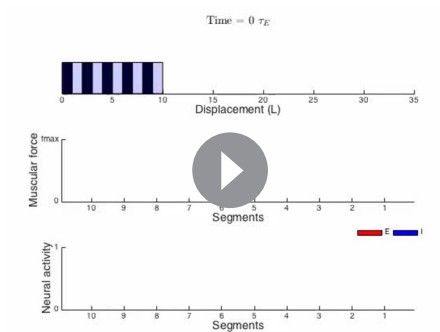

**Video 6.** Crawling simulations of larva with low friction. $F_{\max}/kL = 0.005$.

proprioception, albeit much slower, the demonstration of purely proprioceptive crawling challenges the central role of CPG in crawling locomotion. It will be interesting to see if this finding can be experimentally tested by disrupting intersegmental coupling in the VNC.

Our study suggests that proprioception increases the robustness of crawling by effectively stabilizing segment-to-segment neural coupling. Interpreting this variation in parameters to that due to variations across individuals in a population, we predict that there should be much more variability in crawling metrics among individuals with silenced or weakened proprioception, a scenario amenable to testing using existing genetic tools (**Hughes and Thomas, 2007**; **Inada et al., 2011**).

In the context of body-nervous system coupling, proprioception is necessary for adaptive behavior, but are both neural and proprioceptive intersegmental coupling needed, since crawling is possible with only one of them? Neural coupling may be used to control speed. Indeed, our model predicts that increase in neural coupling leads to higher speeds, even in the presence of proprioception. This suggestion is supported by experiments in stick insects, which show that slow, steady-walking is mainly coordinated by local proprioceptive signals but faster motion is driven by increased neural coupling between legs (**Büschges, 2012**).

Our model makes testable predictions on how crawling should change if body-substrate coupling is modified. In particular, decreasing friction below slippage threshold should lead to a higher speed with larger step sizes but a smaller number of waves per unit time, another testable prediction.

Our minimal model makes a number of assumptions and simplifications. Improvements, some of which are already pointed out, will be necessary to describe forward crawling as more experiments become available. For example, in a very recent experiment (**Itakura et al., 2015**), a new class of pre-motor inhibitory neurons, Glutamatergic Ventro-Lateral Interneurons (GVLIs), were identified (glutamate inhibits larva motor neurons (**Rohrbough and Broadie, 2002**)), which ceased locomotion in the same segment when optogenetically activated. During unperturbed peristaltic wave propagation, GVLIs' activation lagged motor neurons by several segments, suggesting that GVLIs provide a contraction termination signal when the wave reaches anterior segments. In contrast, inhibitory neurons of our model get activated slightly later than the excitatory neurons in the same segment, and receive proprioceptive input only from the same segment. Therefore, our current model does not take into account the role of GLVIs in locomotion. Some other points of improvement could be moving beyond linear mechanics, taking into account different muscles, introducing dynamics for stretch receptors, incorporating observations on PMSI's (**Kohsaka et al., 2014**), including further specialized neuron types and accompanying connectivity profiles in a segment of the VNC, introducing possible long range intersegmental neural and proprioceptive connections, and using more sophisticated parameter fitting procedures.

Our integrated approach suggests generalizations that can move us beyond prograde rectilinear locomotion. Backward crawling, for instance, could be achieved by a separate neural circuit running from anterior-to-posterior, perhaps similar to that in *C. elegans* (**Haspel et al., 2010**), or by introducing anterior to posterior neural coupling between individual segments as in the model of **Gjorgjieva et al. (2013)**. Additional extensions of the body mechanics to account for bending by differential movement of hemisegments, and a bilateral VNC, similar to that in (**Gjorgjieva et al., 2013**), allowing for propagation of neural excitation in opposite directions in different hemisegments will allow us to account for turning, and thus the larger behavioral repertoire of *Drosophila* larvae (**Vogelstein et al., 2014**).

Finally, our model naturally suggests novel biomimetic designs for robotics crawlers. In fact, the interest in soft robots has significantly grown in the last few years thanks to these systems' capability of moving in uncertain environments, a daunting task for traditional rigid robots. For example **Boyle et al., (2013)** presented a proprioceptive-driven articulated crawler based on *C.*

*elegans* morphology and showed that it is able to navigate arenas with unknown obstacles without requiring complex sensory capability. On the other hand, Trimmer and colleagues are developing a soft platform to build artificial crawlers, see for example (*Umedachi et al., 2013*) and (*Kim et al., 2013*) for a review of current attempts. Our model can then be exploited to merge soft robotics and proprioception to create novel biomimetic crawlers with an electromechanical circuit implementing force production and proprioception in a soft gel.

## Acknowledgements

We thank E Heckscher for providing us a movie of crawling in *D. melanogaster* larvae, J Gjorgjieva, H Lacin, S Pulver, A Samuel, B Afonso and A Zarin for discussions, the anonymous reviewers for constructive comments and suggestions, and the Swartz Foundation (CP), and the MacArthur Foundation (LM) for partial financial support.

## Additional information

### Funding

| Funder | Grant reference number | Author |
| --- | --- | --- |
| The Swartz Foundation | Swartz Fellowship | Cengiz Pehlevan |
| John D. and Catherine T. MacArthur Foundation | | L Mahadevan |

The funders had no role in study design, data collection and interpretation, or the decision to submit the work for publication.

### Author contributions

CP, Conception and design, Acquisition of data, Analysis and interpretation of data, Drafting or revising the article; PP, LM, Conception and design, Analysis and interpretation of data, Drafting or revising the article

### Author ORCIDs

Cengiz Pehlevan, http://orcid.org/0000-0001-9767-6063
Paolo Paoletti, http://orcid.org/0000-0001-6131-0377
L Mahadevan, http://orcid.org/0000-0002-5114-0519

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

## Appendix 1

### Crawling simulations in a model with only head-tail neural coupling

Our model has two kinds of long range couplings. First is a mechanical coupling between head and tail, mediated by an infinitely stiff spring, which models the visceral piston-like action of the gut. Such coupling is important for reproducing the experimental observation that head and tail move together and provides a simple, passive mechanism for extending the head segment. Second is the input to $E_{10}$ of tail segment from the stretch receptors and and the excitatory neurons in segment T2. Such input is responsible for reinitiating a crawling wave and sustain crawling gait without a CPG-like, periodic external drive. This kind of coupling needs to be mediated by neural fibers running across the VNC. Experimentally, interneurons that extend their axons across multiple VNC segments have been observed in *Drosophila* larvae (*Schmid et al., 1999*).

Is it possible to build a model without any long range coupling? The head-tail synchrony at the visceral piston phase of crawling requires a means of transmission of timing information. Therefore, some form of a long range coupling is necessary. In this section, we discuss a model of crawling with only neural coupling (*Appendix 1—figure 1A*). Mathematical details of the model are given below, but first we briefly discuss its main properties.

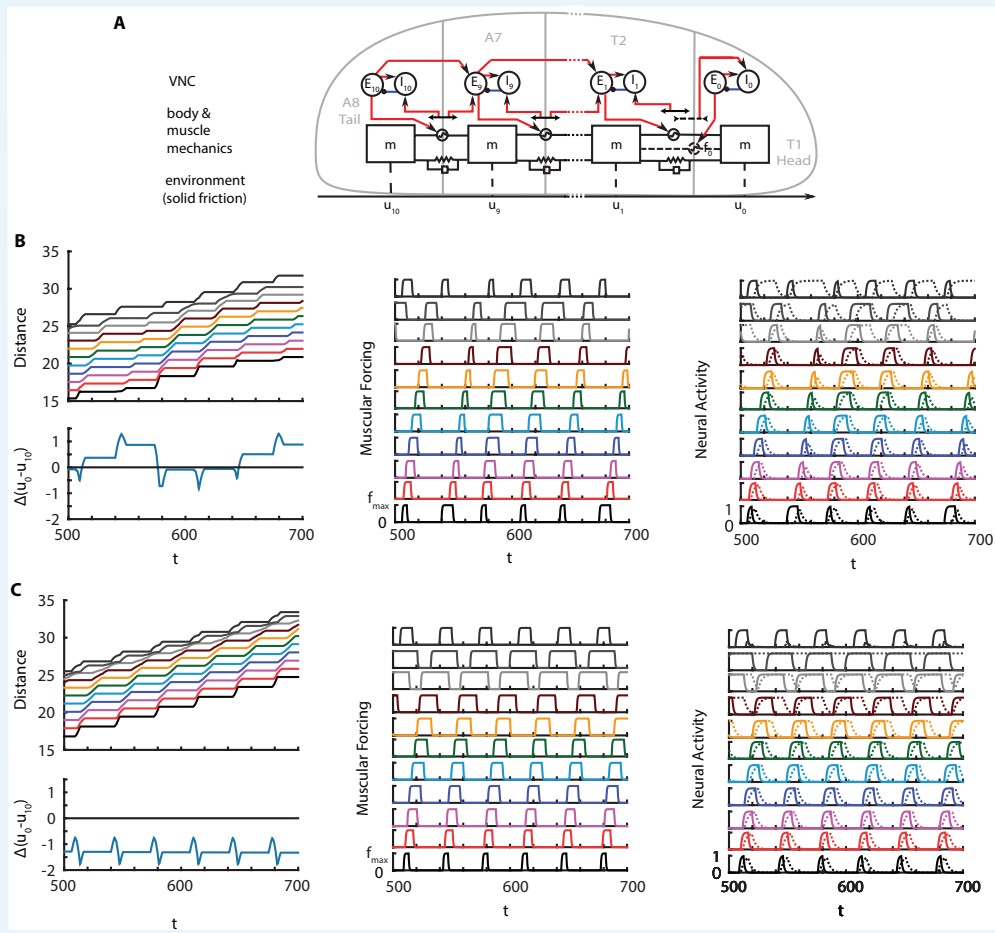

**Appendix 1—figure 1.** Crawling in a model with only head-tail neural coupling. (**A**) Schematic of modified model. (**B**) Perfectly synchronous driving of $E_0$ and $E_{10}$. A full kymograph, muscular

activity and neural activity. Below the kymograph, the changes in total length of the larva is shown. Color code is the same as *Figure 1*. (**C**) Delayed forcing of tail by $0.2\pi$ radians. Same figures as B).

Unfortunately, removal of the head-tail connecting spring from the original model does not ensue a model that produces a crawling gait, as the force that pushes head forward is mediated by the spring. To build a working model, first, an extensile muscular force, $f_0$, should be included at the head segment for it to stretch, which would be implemented by a complex combination of circumferential and longitudinal muscular forces. Second, a neural circuit that controls this new muscle need to be introduced with a neural connection that synchronizes head-tail movements. Such circuit needs to stop the muscular activity when extension is beyond a threshold and therefore needs to be complemented by a receptor that gets activated with stretch, as opposed to receptors that gets activated with contraction in other segments.

Here, we present one such model where head-tail synchronization and crawling cycle reinitialization is achieved by CPG-like pulse inputs to tail and head segments. These pulses could be generated by two separate, synchronized, local oscillators in the VNC, or a common input that feeds both segments. Both cases still require neural fibers that traverse the VNC: in the former case to initialize oscillators simultaneously, and in the latter case to carry the input itself. Such model produces sustained crawling, as illustrated in *Appendix 1—figure 1B*. Not being constrained by a stiff gut, the total length of the larva oscillates around its stationary value, with a periodicity around $150\tau_E$ and changes reaching about 10% of larval length at peaks. The extensile forcing and stretch activated proprioception in the head segment breaks the symmetry between segment and thus causes uneven contraction, muscular forcing and neural activity across segments. An interesting observation is the extended activation of $I_0$ compared to other inhibitory neurons. $I_0$ activates in the begining of the crawling cycle with a proprioceptive input due to head stretching. Such input is alive until the crawling cycle propagates from the tail to T2 and the stretch is dampened.

While this new model looks plausible, it is interesting to note that such model is sensitive to perturbations in synchrony between oscillators. For example, when a delay is introduced to the tail oscillators, which could happen due to delays in propagation of the external start signal, the total length of the crawling larva reduced from its stationary length, as illustrated in *Appendix 1—figure 1C*.

## Details of the model

The dynamical equations are identical to the original model, except that there is an extra excitatory-inhibitory neural population pair at the head segment, labeled by $E_0$ and $I_0$:

$$\tau_E \dot{E}_i = -E_i + \sigma_n[w_{EE}E_i + w_{EI}I_i + h_i^E - \hat{\theta}_E],$$
$$\tau_I \dot{I}_i = -I_i + \sigma_n[w_{IE}E_i + w_{II}I_i + h_i^I - \hat{\theta}_I], \ i = 0, \ldots, 10.$$

Here $h_i^{E,I}$ are again inputs to these population outside the segment and are given by:

$$h_0^E = P_0(t),$$
$$h_i^E = w_{En}E_{i+1} + w_{Ep}\sigma_p[u_{i+1} - u_i - \hat{u}], \ i = 1, \ldots, 9,$$
$$h_{10}^E = P_{10}(t),$$
$$h_0^I = w_{Ip}\sigma_p[u_0 - u_1 - \hat{u}],$$
$$h_i^I = w_{Ip}\sigma_p[u_i - u_{i-1} - \hat{u}], \ i = 1, \ldots, 10.$$

Here, there are a few differences from the original model. First, we show the tonic inputs, $P_0(t)$ and $P_{10}(t)$, that excite $E_0$ and $E_{10}$ explicitly. There is no proprioceptive or neuron-to-neuron input to $E_{10}$ from segment T2. Finally, the proprioceptive input to $I_0$ comes from a contraction detector, rather than a strecth detector.

Displacement of the individual segments $u_i(t)$ are again governed by Newtonian mechanics:

$$0 = k(u_1 - u_0 + L) + c(\dot{u}_1 - \dot{u}_0) - f_1 + f_0 - F_0^f,$$
$$0 = k(u_0 - 2u_1 + u_2) + c(\dot{u}_0 - 2\dot{u}_1 + \dot{u}_2) - f_0 + f_1 - f_2 - F_1^f,$$
$$0 = k(u_{i-1} - 2u_i + u_{i+1}) + c(\dot{u}_{i-1} - 2\dot{u}_i + \dot{u}_{i+1}) + f_i - f_{i+1} - F_i^f, \qquad i = 2, \ldots, 9$$
$$0 = k(u_9 - u_{10} - L) + c(\dot{u}_9 - \dot{u}_{10}) + f_{10} - F_{10}^f.$$

In the absence of 'periodic' boundary conditions, the extensile force, $f_0$, is responsible for extension of the head.

Muscular dynamics are identical to the original model except that there is an extra muscle, $f_0$:

$$\tau_f \dot{f}_i = -f_i + f_{\max} \sigma_f [E_i - \hat{E}], \quad i = 0, \ldots, 10.$$

We again assume that the friction $F_i^f$ drops to zero in a segment when $f_i$ is above a threshold, i.e.

$$F_i^f = F_{\max} \text{sign}(\dot{u}_i) \sigma_F [\hat{f} - f_i], \quad i = 0, \ldots, 10.$$

Parameters of the model are identical to the original model, except $w_{E_p} = w_{I_p} = 2$. The inputs, $P_0(t)$ and $P_{10}(t)$, are rectangular pulses with height 0.7 and witdth 10, with period 33.7. In the model where the tail input is delayed, $P_{10}(t)$ lags $P_0(t)$ by 3.37 time units.

## Appendix 2

### Robustness of simulations to inertial contribution of segments

In our simulations, we introduced a small inertial contribution for the segments that allowed to use explicit numerical integration schemes. Here we show that our results are robust with respect to changes in this parameter.

In *Appendix 2—figure 1*, we re-plot *Figure 6C*, the speed vs. friction force curve, for various values of the dimensionless segmental mass, $m$. The curves differ below $F_{max}/kL \approx m$. Above, they agree. Therefore, it is safe to use an $m$ value much smaller than friction. Similar behavior is observed for all other metrics we looked at. In the main paper, we used the $m = 10^{-5}$ curve which was well below the lowest $F_{max}/kL \approx 3 \times 10^{-4}$ we plotted.

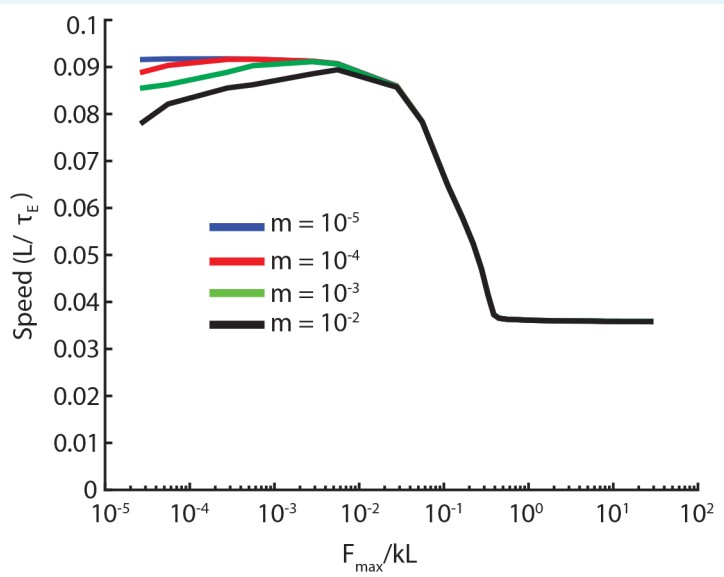

**Appendix 2—figure 1.** Speed vs. friction force for different values of inertia.

