## [Decision Letter]

Thank you for submitting your work entitled "Integrative neuromechanics of crawling in *D. melanogaster* larvae" for consideration by *eLife*. Your article has been reviewed by three peer reviewers, and the evaluation has been overseen by Ronald L. Calabrese as the Reviewing Editor and K VijayRaghavan as the Senior Editor.

The reviewers have discussed the reviews with one another and the Reviewing editor has drafted this decision to help you prepare a revised submission.

Summary:

The neuro-mechanics underlying peristalsis in the *Drosophila* larva offers an excellent opportunity to understand the link between a body, its nervous system and the environment surrounding it. In recent years, the neural network underlying crawling in the larva has been investigated by numerous labs. Theorists have started integrating experimental data in quantitative models, as was done in pioneering work by Gjorgjieva et al. 2013. The present manuscript offers a detailed model of the dynamics produced by the neuro-mechanical coupling of larval segments. The model is minimally complex, which makes it elegant and powerful. Using this model, the authors explore the influence of proprioceptive inputs and body-substrate friction on quantitative aspects crawling behavior. Many behavioral observations are reproduced. Remarkably, the authors propose a set of experimentally testable predictions. The strength of the manuscript is the novelty and the elegance of the model, its ability to account for a number of experimental observed phenomena, and its clear experimental predictions.

Essential revisions:

The major weaknesses pointed out in the expert reviews and which require revision are:

1) At the moment, the model is presented for applied mathematicians but not biologists. Given that this paper targets an audience of biologists, an effort should be made to create a more intuitive understanding of the math behind the model. To help this issue, we suggest that the authors add a model overview section, preceding the current modeling section, that explains the physics and the biology of the model in a way that is accessible to a biologist, but without equations. Such a section would have to be written so that a reader who wants to struggle through the equations will have read the overview and therefore have a good intuitive understanding of what the model consists of/model assumptions etc., whereas a reader who chooses to skip the technical section can still fully follow the Results and Discussion of the paper. Particularly indicate how the parameters of the model pertain to specific interactions perhaps in reference to an expanded Figure 1. Given that the model contains a large number of parameters more effort must be made to explain how the different parameters were fitted to the model. It is important to discuss the limitations inherent in the assumptions of the model (i.e. obvious discrepancies with/simplifications of the biological process) and in the future how might these limitations be overcome?

2) In 2013, Gjorgjieva et al. proposed a model similar to that presented by the authors. Although the conclusions of both studies do not fully overlap, the authors should discuss more thoroughly how their findings compare to those of Gjorgjieva et al. One striking difference is related to the necessary involvement of a CPG to drive peristalsis. The authors demonstrate that peristalsis can emerge from a pure neuro-mechanical coupling between neighboring segments. This result should be highlighted in the context of the CPG-centric view prevailing in the field. Perhaps part of the discussion found in supplementary information should be moved to the main text? The authors should also emphasize the fundamental differences between a CPG-based and pure neuro-mechanical coupling model.

3) Currently, the model does not integrate the results of two recent studies from the lab of Aki Nose:

A) Kohsaka H, Takasu E, Morimoto T, Nose A. A group of segmental premotor interneurons regulates the speed of axial locomotion in *Drosophila* larvae. Curr Biol. 2014 Nov 17;24(22):2632-42. doi: 10.1016/j.cub.2014.09.026. Epub 2014 Oct 16. PubMed PMID: 25438948.

B) Itakura Y, Kohsaka H, Ohyama T, Zlatic M, Pulver SR, Nose A. Identification of Inhibitory Premotor Interneurons Activated at a Late Phase in a Motor Cycle during *Drosophila* Larval Locomotion. PLoS One. 2015 Sep 3;10(9):e0136660. doi: 10.1371/journal.pone.0136660. eCollection 2015. PubMed PMID: 26335437; PubMedCentral PMCID: PMC4559423.

These papers should be considered. See B. and D. below for more detail.

Below are excerpts from the expert reviews that expand on these needed revisions.

A) The model is simple and elegant. Necessarily, it includes a large number of assumptions and simplifications. These are in general clearly stated, but some of these lack justification (and references to the neurobiological evidence). For example, the proprioceptive coupling is only from the nearby segment (contraction of segment i+1 feeds into the neural activation of segment i). Furthermore, such coupling is subject to a highly nonlinear activation function (effectively a threshold), and is fast (the very strong nonlinearity is somewhat reminiscent of other models, e.g. in *C. elegans*, Bryden 2008, Boyle 2012) begging the question is such strong nonlinearity strictly necessary for the model to work and does this render the dynamics of each segment as effectively binary, up to the capacitive filtering?

B) The model assumes one excitatory and one inhibitory neuronal cell type. Recent literature has revealed more specialized cell types, e.g. excitatory PMSIs (Kohsaka et al., Curr Biol. 2014;24(22):2632-42.) and inhibitory GVLIs (Itakura et al. 2015, PLoS ONE 10(9):e0136660), with both classes implicated in peristaltic locomotion, and at least the inhibitory class also possibly consistent with the existing "mission accomplished" intuition. Is this more recent literature consistent with the model? Would it be possible to add a schematic with relevant cell types and lay out the reasoning of how these may be simplified/collapsed to the model presented?

C) The mathematical model builds on a 2014 paper by two of the authors (Paoletti and L Mahadevan, Proc. R. Soc. B, 2014). While this is stated, much more detail needs to be given to explain what elements of the previous model framework are built on, and what elements of the current model and results are novel (e.g. are the results in sections 3.2.1 new?). Specifically, the previous model seems to focus on the passive and active body model and a minimal control system consisting of nearest-neighbor inter-segmental proprioceptive and diffusive coupling. There, the rhythmic pattern was generated by an oscillator in the head. This model now adds neuronal detail (pattern generating neuronal populations in each segment along the VNC). Thus in this model, some patterns can be propagated along the body even in the absence of ascending (proprioceptive) or descending (brain) control, and the proprioception is then superimposed (with a demonstrated stabilizing effect). This offers an excellent framework to study the interplay between central and peripheral control in this system. It would be helpful to the readers if any differences from the previous model (regarding the biomechanics/proprioception) and head oscillator were highlighted and justified. Where there are none, this should also be stated.

Background and literature search – I was surprised the very relevant work on leech proprioception was not cited, e.g. Cang and Friesen (2002), Model for Intersegmental Coordination of Leech Swimming: Central and Sensory Mechanisms. J. Neurophys. 87(6): 2760-2769.

D) The inhibitory interneurons described in the last paragraph of section 2.1 should be related to the population of local interneurons neurons described by Kohsaka et al. 2014: the PMSIs. The same should be done for the glutamatergic interneuron GVLIs described in Itakura et al. 2015. The predictions of the model should be compared to the activity patterns reported for the PMSIs and GVLIs. The authors will note that the dynamics of GVLIs does not appear to be consistent with interactions that purely involves neighboring segments (the GVLIs lag behind the motor neurons by several segments).

Related to this point, Gjorgjieva et al. 2013 assumed a coupling of ipsilateral excitatory and inhibitory interneurons (nearest-neighbor coupling through inhibitory interneurons), which is different from that used by the authors (nearest-neighbor coupling through excitatory interneurons). Should one favor one connectivity pattern over the other one? Could the functional implications of these differences be commented on?

E) In the last paragraph of the section 3.1.2: the authors make an equivalence between the acute loss-of-function achieved through the expression of halo-rhodopsin experiments in the motorneurons (see Inada et al., 2011) and an acute increase in the activity of the inhibitory interneurons. In my view, this mathematical description of the loss-of-function is incorrect. The effect of halo-rhodopsin should be described as a dramatic reduction in the activity of the excitatory interneurons (possibly through a large increase in the value of Ei in equation 1). This perturbation might have different consequences on the network dynamics. In particular, it might explain why the authors still predict muscular contractions upon activation of halo-rhodospin, which seems incompatible with the inhibition of the motorneurons.

F) At several occasions, the length of larvae is incorrectly approximated. In the second paragraph of the Introduction, third instar larvae are assumed to be 8 mm long. I was wondering where this estimate comes from. In my experience, third instar larvae are typically 4 mm long, with a maximum length of 5 mm. In section 3.1.1, the length of the larva is assumed to be 0.7mm. Did the authors intend to write 7mm? In any case, this estimate is justified by referring to the Figure 1 in Inada et al., 2011, but the scale bar on the figure of reference shows that the larva is not larger than 4mm. The authors should correct these dimension and update their parameter estimates.

G) In the Discussion, the authors should consider aspects of the biological phenomenology that have been neglected in their model. (1) The coupling of left and right hemi-segments. Gjorgjieva et al. 2013 have discussed that the contralateral coupling of excitatory and inhibitory interneurons might be important to produce robust propagation of peristaltic waves. How would such coupling affect the present model? (2) Can the present model account for the propagation of backward waves during backing-up movements?

H) It is not quite clear to me how the approach of Mahadevan and his colleagues differs from other modeling studies investigating crawling in *D. melanogaster* larvae or *C. elegans* or other worms or snakes. The authors state in the Introduction that recent studies on *C. elegans* (Wen et al., 2012; Boyle, Berri and Cohen, 2012) have already focused on local sensory feedback and proprioception and show that this suffices to modulate the locomotory pattern and explain gait transitions. Boyle and his colleagues have also shown that proprioception stabilizes crawling and makes it more robust against external perturbation. Furthermore, old studies by Niebur and Erdös in the 90s have already started to emphasize the role of sensory feedback in gait generation. Or, the works on crawling in leeches by Friesen and colleagues. I think it would be really helpful for the reader if the authors could include a paragraph into the Discussion stating how their approach differs from existing ones and what the specific advantages of their approach are.

I) The second point which was not quite clear to me concerns the coupling structure (neural and proprioceptive) that the authors used in their simulations. Which parts of this structure are based on neurobiological findings in D. melanogaster larvae and which are based on physiologically reasonable assumptions, because they are known to exist in other biological systems? In line with this, the question whether different neural coupling structures, e.g. including anterior to posterior couplings between the individual segments will interfere with the results, arises.

In case that not much is known about the neurobiology of the coupling structure and that it is mainly based on assumptions, did the authors test and compare different central/neural coupling structures and their outputs?

---

## [Author Response]

Essential revisions:

The major weaknesses pointed out in the expert reviews and which require revision are:

1) At the moment, the model is presented for applied mathematicians but not biologists. Given that this paper targets an audience of biologists, an effort should be made to create a more intuitive understanding of the math behind the model. To help this issue, we suggest that the authors add a model overview section, preceding the current modeling section, that explains the physics and the biology of the model in a way that is accessible to a biologist, but without equations. Such a section would have to be written so that a reader who wants to struggle through the equations will have read the overview and therefore have a good intuitive understanding of what the model consists of/model assumptions etc., whereas a reader who chooses to skip the technical section can still fully follow the Results and Discussion of the paper. Particularly indicate how the parameters of the model pertain to specific interactions perhaps in reference to an expanded Figure 1. Given that the model contains a large number of parameters more effort must be made to explain how the different parameters were fitted to the model. It is important to discuss the limitations inherent in the assumptions of the model (i.e. obvious discrepancies with/simplifications of the biological process) and in the future how might these limitations be overcome?

We thank the reviewers for encouraging us to make our presentation more accessible to biologists, who are our intended audience. As suggested, we introduced a new section (section 2.1) titled Overview, which aims to give an intuitive understanding of the model. The reader should be able to read this section, skip the mathematical details, and continue with the Results. The overview section is accompanied by an expanded Figure 1 new panel zooms in a single segment and describes the important features and parameters of the model. In choosing the parameters we did not follow and automated approach but followed a few heuristics, which we now discuss in section 2.5. In section 2, we were already careful in pointing to the limitations of the model. Now this is accompanied by a much-expanded Discussion section where shortcomings of our model and possible ways to extend it are discussed.

2) In 2013, Gjorgjieva et al. proposed a model similar to that presented by the authors. Although the conclusions of both studies do not fully overlap, the authors should discuss more thoroughly how their findings compare to those of Gjorgjieva et al. One striking difference is related to the necessary involvement of a CPG to drive peristalsis. The authors demonstrate that peristalsis can emerge from a pure neuro-mechanical coupling between neighboring segments. This result should be highlighted in the context of the CPG-centric view prevailing in the field. Perhaps part of the discussion found in supplementary information should be moved to the main text? The authors should also emphasize the fundamental differences between a CPG-based and pure neuro-mechanical coupling model.

We thank the reviewer for bringing up these important issues. While the model of Gjorgjieva et al. was purely a neural model, some of their findings indeed overlap with ours. We added a paragraph in model section 2.2 highlighting the similarities and differences between our model and that of Gjorgjieva et al. Further, we added two more references to their paper in section 3.2.1 highlighting similarities in our results.

It is true that our model can produce crawling with pure proprioceptive coupling between segments. However, because the larva can crawl with silenced proprioception, see e.g. Hughes & Thomas 2007, we do not believe such scenario is a viable model for larval crawling. Nevertheless, following the reviewer’s suggestion, we added a paragraph in Discussion highlighting this finding. We should also remind the reviewer that the possibility of sustained crawling with purely proprioceptive coupling was demonstrated and discussed in a model by two of the authors before (Paoletti and Mahadevan, 2014). Finally, the discussion in the supplementary information is not related a pure neuro-mechanical model, therefore we chose to keep it as it is.

*3) Currently, the model does not integrate the results of two recent studies from the lab of Aki Nose:*

*A) Kohsaka H, Takasu E, Morimoto T, Nose A. A group of segmental premotor interneurons regulates the speed of axial locomotion in Drosophila larvae. Curr Biol. 2014 Nov 17;24(22):2632-42. doi: 10.1016/j.cub.2014.09.026. Epub 2014 Oct 16. PubMed PMID: 25438948.*

B) Itakura Y, Kohsaka H, Ohyama T, Zlatic M, Pulver SR, Nose A. Identification of Inhibitory Premotor Interneurons Activated at a Late Phase in a Motor Cycle during Drosophila Larval Locomotion. PLoS One. 2015 Sep 3;10(9):e0136660. doi: 10.1371/journal.pone.0136660. eCollection 2015. PubMed PMID: 26335437; PubMedCentral PMCID: PMC4559423.

These papers should be considered. See B. and D. below for more detail.

We thank the reviewers for bringing these papers to our attention. Indeed, the inhibitory interneurons described in the first reference, PMSIs, seem to match some aspects of the inhibitory neurons in our model, i.e. that they get activated slightly after the motor neurons in the same segment and that their optogenetic excitation leads to a local arresting of the peristaltic wave. Now, we added a simulation showing this latter point, Figure 3, and expanded the Results section 3.1.2 to discuss this simulation. Kohsaka et al. report on other results that are not consistent with our model, which we discuss in detail in section 3.1.2.

On the other hand, the activation of inhibitory interneurons described in the second reference, GLVIs, lag the motor neurons by 2-3 segments. Itakura et al. speculate that GVLIs provide a contraction termination signal when the wave reaches anterior segments. In contrast, our inhibitory neurons get only input from only the same segment. Therefore, our current model does not take into account the role of GLVIs in locomotion. We discuss this shortcoming of our model in a new paragraph in the Discussion section.

*Below are excerpts from the expert reviews that expand on these needed revisions.*

A) The model is simple and elegant. Necessarily, it includes a large number of assumptions and simplifications. These are in general clearly stated, but some of these lack justification (and references to the neurobiological evidence). For example, the proprioceptive coupling is only from the nearby segment (contraction of segment i+1 feeds into the neural activation of segment i). Furthermore, such coupling is subject to a highly nonlinear activation function (effectively a threshold), and is fast (the very strong nonlinearity is somewhat reminiscent of other models, e.g. in C. elegans, Bryden 2008, Boyle 2012) begging the question is such strong nonlinearity strictly necessary for the model to work and does this render the dynamics of each segment as effectively binary, up to the capacitive filtering?

We are pleased to see that the reviewer finds the model elegant. We did do a large number of assumptions and simplifications, and we tried our best to point to biological literature when for evidence when it exists. However, there are many unknowns about how the larva crawls. When faced with an unknown, we assumed the minimal mechanism. The reviewer points to two such assumptions:

1) The proprioceptive coupling is only from the nearby segment: It is true that proprioceptive coupling could extend multiple segments, however in the lack of experimental data, we chose the minimal coupling, which is the nearest neighbor coupling.

2) The nonlinearity of the relationship between strain and proprioceptive signal: We want to make two points about this. 1) No characterization of a strain-output relationship of a proprioceptor is published in the larva. The sigmoidal relationship we used between strain and proprioceptive signal is not merely chosen for convenience: A sigmoid is a strong nonlinearity, but it is also a plausible nonlinearity. One would expect neurons detecting segmental contraction to be silent for low contraction, and saturate after some point. 2) It is true that the gain of the sigmoid in our model is large (1000), and it renders the proprioceptive signal to be binary. We can lower this gain from 1000 to 5 (which is still pretty steep), with little qualitative and quantitative change in sustained crawling. Below 5, the model stops producing sustained crawling. We do not want to draw strong conclusions from this observation and “predict” the necessity of a strong nonlinearity for crawling, because there are many other parameters to adjust which may allow gains below 5 possible.

The dynamics of each segment being effectively binary, up to the capacitive filtering (by which we believe the reviewer means the filtering due to springs in equation 3) is approximately true, because there are other time constants in play. The proprioceptive signal passes through neurons and muscles, each of which has its own time constants and nonlinearities, before feeding into the dynamical equations for segments (eq. 3). However, it is obvious, and we admit, that the dynamics described here is simple and would need to be modified to explain more complicated phenomena.

B) The model assumes one excitatory and one inhibitory neuronal cell type. Recent literature has revealed more specialized cell types, e.g. excitatory PMSIs (Kohsaka et al., Curr Biol. 2014;24(22):2632-42.) and inhibitory GVLIs (Itakura et al. 2015, PLoS ONE 10(9):e0136660), with both classes implicated in peristaltic locomotion, and at least the inhibitory class also possibly consistent with the existing "mission accomplished" intuition. Is this more recent literature consistent with the model? Would it be possible to add a schematic with relevant cell types and lay out the reasoning of how these may be simplified/collapsed to the model presented?

We thank the reviewer for bringing these papers to our attention. The results reported on PMSIs (which are inhibitory) do match some aspects of the inhibitory neurons in our model, i.e. that they get activated slightly after the motor neurons in the same segment and that their optogenetic excitation leads to a local arresting of the peristaltic wave. A simulation of such local arrest is added to the Results section 3.1.2 (Figure 3). Kohsaka et al. report on other results that are not consistent with our model, which we discuss in detail in a new paragraph again in section 3.1.2.

Activation of GLVIs on the other hand lag the motor neurons by 2-3 segments. We therefore believe that our current model does not take into account the role of GLVIs in locomotion. We discuss this shortcoming of our model in a new paragraph in the Discussion section. Since our model does not include GLVIs, we did not add the schematic that the reviewer requested.

*C) The mathematical model builds on a 2014 paper by two of the authors (Paoletti and L Mahadevan, Proc. R. Soc. B, 2014). While this is stated, much more detail needs to be given to explain what elements of the previous model framework are built on, and what elements of the current model and results are novel (e.g. are the results in sections 3.2.1 new?). Specifically, the previous model seems to focus on the passive and active body model and a minimal control system consisting of nearest-neighbor inter-segmental proprioceptive and diffusive coupling. There, the rhythmic pattern was generated by an oscillator in the head. This model now adds neuronal detail (pattern generating neuronal populations in each segment along the VNC). Thus in this model, some patterns can be propagated along the body even in the absence of ascending (proprioceptive) or descending (brain) control, and the proprioception is then superimposed (with a demonstrated stabilizing effect). This offers an excellent framework to study the interplay between central and peripheral control in this system. It would be helpful to the readers if any differences from the previous model (regarding the biomechanics/proprioception) and head oscillator were highlighted and justified. Where there are none, this should also be stated.*

Background and literature search – I was surprised the very relevant work on leech proprioception was not cited, e.g. Cang and Friesen (2002), Model for Intersegmental Coordination of Leech Swimming: Central and Sensory Mechanisms. J. Neurophys. 87(6): 2760-2769.

As the reviewer points out, and we state in the paper, our model builds on a 2014 paper by 2 of the authors (Paoletti and Mahadevan, 2014), but has many differences from it. Most importantly, the previous model was built thinking of a “generic” crawler to prove the point that crawling without a CPG is possible. Therefore, it lacked a neural coupling between segments. Besides, the neural dynamics within segments were approximated with an oscillator. The current paper aims to model larval crawling: It has neural coupling between segments because the larva is able to crawl without proprioception (Hughes and Thomas, 2007). Other main differences are: 1) Our current model does not need a start signal at the beginning of each wave as in the old model, but restarts a wave by the neural coupling between T2 and A8, 2) In the current model, friction is set to zero when muscular forces exceed a threshold, modeling lifting of the body from the ground. In the old model, friction changed by direction of motion. 3) The dynamics of neurons in the old model (phase oscillators) were different than the current one and there was only one population of neurons. 4) The proprioception only activated the next segment but did not inhibit the current segment. 5) The old model was actually numerically solved in the continuum limit assuming a large number of segments, whereas here we solve the model with discrete segments. These differences are now pointed in the mathematical model section.

Regarding novelty of our results, we claim that all are novel, except that the model is able to crawl with purely proprioceptive coupling, which is just one of the results reported in section 3.2.1. We inserted a citation to the old model, in addition to the existing one in section 2, when we discuss purely proprioceptive crawling in section 3.2.1. The rest of our results, an overwhelming majority, are novel because: 1) They involve features that did not exist in the previous model: perturbations of excitatory and inhibitory neurons (section 3.1.2), crawling with purely neural coupling (section 3.1.3), manipulations of neural coupling (section 3.2.1). 2) They report results that the previous model did not address: resemblance to *D. melanogaster* larva crawling (section 3.1.1) and role of friction (section 3.2.2).

We are pleased to see that the reviewer agrees that our model offers an excellent framework to study the interplay between central and peripheral control in this system. We did our best to exploit this feature in this paper.

We apologize for missing the references to the leech proprioception. We added references to it in the Introduction.

*D) The inhibitory interneurons described in the last paragraph of section 2.1 should be related to the population of local interneurons neurons described by Kohsaka et al. 2014: the PMSIs. The same should be done for the glutamatergic interneuron GVLIs described in Itakura et al. 2015. The predictions of the model should be compared to the activity patterns reported for the PMSIs and GVLIs. The authors will note that the dynamics of GVLIs does not appear to be consistent with interactions that purely involves neighboring segments (the GVLIs lag behind the motor neurons by several segments).*

Related to this point, Gjorgjieva et al. 2013 assumed a coupling of ipsilateral excitatory and inhibitory interneurons (nearest-neighbor coupling through inhibitory interneurons), which is different from that used by the authors (nearest-neighbor coupling through excitatory interneurons). Should one favor one connectivity pattern over the other one? Could the functional implications of these differences be commented on?

We thank the reviewer for bringing these papers to our attention. We now relate the inhibitory population of our model to PMSIs in the Results section. Experimentally, PMSIs get activated slightly after the motor neurons in the same segment and their optogenetic excitation leads to a local arresting of the peristaltic wave. A simulation of such local arrest due to extra excitation of our inhibitory neurons is added to the Results section 3.1.2 (Figure 3). Kohsaka et al. report on other results that are not consistent with our model, which we discuss in detail again in section 3.1.2. As the reviewer correctly points, activation of GLVIs lag the motor neurons by 2-3 segments and hence the inhibitory neurons in our model does not describe GLVIs. In a new paragraph in the Discussion section, we discuss this inconsistency.

Gjorgieva et al. assumed two kinds of bidirectional intersegmental couplings: 1) connections between the excitatory populations of the neighboring segments and 2) connections from the inhibitory populations to the neighboring excitatory populations. Our model only has the first type and it is unidirectional. Gjorgieva et al. comment that the function of the second type of connections is “to terminate activity in the previously active segment and ensure unidirectional wave propagation.” In our model, we were able to achieve termination of activity in the previously active segment by the activity of inhibitory neurons in that segment. Unidirectional wave propagation was built into the unidirectional connectivity in the intersegmental coupling. Therefore, our model did not need connections between inhibitory and excitatory neurons in neighboring segments and hence our choice of not including them. Favoring one coupling versus the other one will ultimately depend on experiments and currently we do not know of an experimental result that can distinguish between the two cases. In a new paragraph in section 2.2, we point to the differences between the results of Gjorgieva et al. and ours.

E) In the last paragraph of the section 3.1.2: the authors make an equivalence between the acute loss-of-function achieved through the expression of halo-rhodopsin experiments in the motorneurons (see Inada et al., 2011) and an acute increase in the activity of the inhibitory interneurons. In my view, this mathematical description of the loss-of-function is incorrect. The effect of halo-rhodopsin should be described as a dramatic reduction in the activity of the excitatory interneurons (possibly through a large increase in the value of Ei in equation 1). This perturbation might have different consequences on the network dynamics. In particular, it might explain why the authors still predict muscular contractions upon activation of halo-rhodospin, which seems incompatible with the inhibition of the motorneurons.

The reviewer is completely right in that “the effect of halo-rhodopsin should be described as a dramatic reduction in the activity of the excitatory interneurons”. This is exactly what we did in the model, however our wording was mistakenly suggests that we increased the activity of inhibitory interneurons. What was done was reducing the activity of excitatory population, by setting it to zero. This current did *not* come from an increase in inhibitory activity (see Figure 3). We put in some more explanation in the paragraph to make this point clear.

*F) At several occasions, the length of larvae is incorrectly approximated. In the second paragraph of the Introduction, third instar larvae are assumed to be 8 mm long. I was wondering where this estimate comes from. In my experience, third instar larvae are typically 4 mm long, with a maximum length of 5 mm. In section 3.1.1, the length of the larva is assumed to be 0.7mm. Did the authors intend to write 7mm? In any case, this estimate is justified by referring to the Figure 1 in Inada et al., 2011, but the scale bar on the figure of reference shows that the larva is not larger than 4mm. The authors should correct these dimension and update their parameter estimates.*

Many thanks for catching this one! Indeed, we made a mistake in our estimates. We changed our third instar larvae length assumption to 4mm. In line 229, we assumed L = 0.7 mm, which meant the length of the larva to be 10L = 7mm. We now changed it the assumption to 10L = 4mm. In addition, we rewrote the paragraph where our estimates are given, to avoid confusions. The section now reads:

“Our simulations show that the larva produces about 0.038 waves per unit time associated with the relaxation of the excitatory neurons 𝜏", where waves start when the tail moves off the ground, leading to a larval speed ≈ 0.036L/𝜏". […] This in perfect agreement with the range reported in (Heckscher, Lockery and Doe, 2012): ≈ 39.5-126.5 𝜇m/sec.”

G) In the Discussion, the authors should consider aspects of the biological phenomenology that have been neglected in their model. (1) The coupling of left and right hemi-segments. Gjorgjieva et al. 2013 have discussed that the contralateral coupling of excitatory and inhibitory interneurons might be important to produce robust propagation of peristaltic waves. How would such coupling affect the present model? (2) Can the present model account for the propagation of backward waves during backing-up movements?

Both issues brought up here are indeed phenomenology neglected in our model. We acknowledge both points (and more) in the penultimate paragraph of our Discussion. To answer in more detail:

1) Addressing the effects of contralateral coupling of excitatory and inhibitory neurons requires substantial additions to our model. In addition to building a two-sided model of VNC as in Gjorgieva et al., we need to introduce a two-sided mechanical model for body hemisegments. It is hard for us to say how the contralateral coupling would affect such model, without actually building it. We intend to do so in future work.

2) Repeating what is said in the paper, “Backward crawling, for instance, could be achieved by a separate neural circuit running from anterior-to-posterior, perhaps similar to that in C. elegans (Haspel, Donovan and Hart, 2010).” Yet another mechanism could be to introduce anterior to posterior couplings between the individual segments as in (Gjorgjieva et al. 2013) to allow propagation of activity in VNC in the opposite direction. We added this possibility to the sentence above, “…or by introducing anterior to posterior neural coupling between individual segments as in the model of (Gjorgjieva et al., 2013).”

H) It is not quite clear to me how the approach of Mahadevan and his colleagues differs from other modeling studies investigating crawling in D. melanogaster larvae or C. elegans or other worms or snakes. The authors state in the Introduction that recent studies on C. elegans (Wen et al., 2012; Boyle, Berri and Cohen, 2012) have already focused on local sensory feedback and proprioception and show that this suffices to modulate the locomotory pattern and explain gait transitions. Boyle and his colleagues have also shown that proprioception stabilizes crawling and makes it more robust against external perturbation. Furthermore, old studies by Niebur and Erdös in the 90s have already started to emphasize the role of sensory feedback in gait generation. Or, the works on crawling in leeches by Friesen and colleagues. I think it would be really helpful for the reader if the authors could include a paragraph into the Discussion stating how their approach differs from existing ones and what the specific advantages of their approach are.

The reviewer is correct in pointing that the role of sensory feedback in locomotion has been investigated before. However, our approach has novelty. First, we point that the gait we study is rectilinear crawling, which is different than the gaits studied before. As the reviewer would appreciate, it is not guaranteed that results obtained from other gaits are going to automatically generalize to all gaits, due to differences in mechanics, neural control and the interactions with the environment. The model that we present for crawling is novel and simple. Second, we study the crawling gait in *Drosophila melanogaster* larvae. As we discuss in the introduction, D. *melanogaster* is an excellent system because of especially the genetic and optogenetic tools that can be used to perturb neural and proprioceptive systems and see their effects on crawling. In the paper we reproduce many of these experiments. We don’t think any of the models cited by the reviewer have been subject to such detailed testing, except maybe that of Boyle et al. Further, we emphasize and predict how changes in friction can affect the gait, something that has not been considered in *D. melanogaster* literature. To summarize, the strength of our approach comes from the model system and the gait we study: we propose a simple and novel model for the crawling gait, our model reproduces many of perturbation experiments in *D. melanogaster* larvae and it produces clear, testable predictions.

We added the following sentences to the end of the first paragraph to address these points: “However, such attempts were mostly aimed at quantitatively replicating organisms' behaviors, without any rigorous experimental validation. Moreover, the complexity of the studied organisms prevents exploiting such models to make testable predictions that can be used to shed light on the biological mechanisms regulating locomotion.”

*I) The second point which was not quite clear to me concerns the coupling structure (neural and proprioceptive) that the authors used in their simulations. Which parts of this structure are based on neurobiological findings in D. melanogaster larvae and which are based on physiologically reasonable assumptions, because they are known to exist in other biological systems? In line with this, the question whether different neural coupling structures, e.g. including anterior to posterior couplings between the individual segments will interfere with the results, arises.*

*In case that not much is known about the neurobiology of the coupling structure and that it is mainly based on assumptions, did the authors test and compare different central/neural coupling structures and their outputs?*

There does not exist an anatomical study where the coupling structure in the VNC is mapped out. Therefore, there is no reason to choose one connectivity over the other, as long as one can reproduce experimental results. Therefore, as we mention a few times in the paper, we chose to use a minimal model that is able to reproduce experimental results and further able to make clear, testable predictions.

Different coupling structures may interfere with results, one of our contributions is to identify a coupling structure that can reproduce results of experiments where neural and proprioceptive systems are perturbed. This does not mean that our results are only specific to the particular coupling we used. For example, our results are consistent with that of (Gjorgjieva et al. 2013), where they considered a purely neural model with anterior to posterior couplings between segments (the scenario that the reviewer is inquiring about). They found that “sensory feedback improves wave propagation” as we also found, or that increasing excitatory coupling between segments makes the waves propagate faster as we also observed. The overlap between our results and that of Gjorgjieva et al. makes us believe that including anterior to posterior couplings between the individual segments will not interfere with our results. We added references to the overlap between and our results and (Gjorgjieva et al. 2013) in the section 3.2.1.

The space of all possible coupling structures is vast, and the reviewer would hopefully agree that it is an unreasonably complicated task to test and compare all such structures. Our aim here, after all, is to present a model, as minimal as it could be, to reproduce experiments and make testable predictions, as opposed to ruling out all implausible ones.

Finally, in the supplementary information, we do present an alternative coupling structure, where head and tail are driven by separate neural controls and are synchronized by an external signal. We show that such model can also produce sustained crawling gait. The reason we chose to implement this scenario was to point that the nature of head-tail synchrony, whether it is mechanical as in the main text or neural as in the supplementary information, does not change our results.